# Towards Generalizable PDE Dynamics Forecasting via Physics-Guided Invariant Learning

**Siyang Li[1], Yize Chen[2], Yan Guo[3], Ming Huang[3], Hui Xiong[1]***
[1]Hong Kong University of Science and Technology (Guangzhou)
[2]University of Alberta, [3]Alibaba Cloud
`sli572@connect.hkust-gz.edu.cn, yize.chen@ualberta.ca,`
`mingqian.hm@alibaba-inc.com, qingjian.gy@alibaba-inc.com`
`xionghui@hkust-gz.edu.cn`

## Abstract

Advanced deep learning-based approaches have been actively applied to forecast the spatiotemporal physical dynamics governed by partial differential equations (PDEs), which acts as a critical procedure in tackling many science and engineering problems. As real-world physical environments like PDE system parameters are always capricious, how to generalize across unseen out-of-distribution (OOD) forecasting scenarios using limited training data is of great importance. To bridge this barrier, existing methods focus on discovering domain-generalizable representations across various PDE dynamics trajectories. However, their zero-shot OOD generalization capability remains deficient, since extra test-time samples for domain-specific adaptation are still required. This is because the fundamental physical invariance in PDE dynamical systems are yet to be investigated or integrated. To this end, we first explicitly define a two-fold PDE invariance principle, which points out that ingredient operators and their composition relationships remain invariant across different domains and PDE system evolution. Next, to capture this two-fold PDE invariance, we propose a physics-guided invariant learning method termed **iMOOE**, featuring an Invariance-aligned Mixture Of Operator Expert architecture and a frequency-enriched invariant learning objective. Extensive experiments across simulated benchmarks and real-world applications validate iMOOE's superior in-distribution performance and zero-shot generalization capabilities on diverse OOD forecasting scenarios. The code is be publicly accessible at `https://github.com/LSY-Cython/iMOOE`.

## 1 Introduction

Reasoning physical dynamics governed by partial differential equations (PDEs) is essential for a wide range of science and engineering applications, such as meteorological prediction (Pathak et al., 2022), battery design (Wang et al., 2024a), chemical synthesis (Gao & Günnemann, 2024) and electromagnetic simulation (Huang et al., 2022). As real-world PDE dynamical systems are always complex, ever-changing and even unknown, it is difficult for traditional numerical methods to explicitly discover the physical law, which requires intensive expert knowledge and computation resources. To this end, physics-informed deep learning (Yu & Wang, 2024; Li et al., 2024c) are applied to identify unknown PDE dynamics and speed up calculation. For instance, neural operators (Kovachki et al., 2023; Li et al., 2023c) are developed to discover the underlying PDE law based on observed trajectories and geometries. Score-based generative models (Li et al., 2024b; Shysheya et al., 2024) are employed to reconstruct the full physical field from sparse measurements. Despite these success, the zero-shot out-of-distribution (OOD) generalization performance of PDE dynamics learning remains underexplored. It is crucial to achieve accurate zero-shot PDE forecasting on unseen OOD scenarios without additional adaptation. It can obviate test-time retraining burden and accelerate various PDE system design and control problems (Hao et al., 2022).

---

*Corresponding author.

To tackle OOD challenges in PDE dynamics learning, existing works focus on learning domain-generalizable representations from multi-domain PDE dynamics. Such domain can be governed by variable physical parameters in PDE systems (Cho et al., 2024). This line of research can be categorized into three classes. First, domain-aware meta-learning (Zintgraf et al., 2019) is leveraged to empower PDE forecasting models with fast adaptation ability to test domains (Wang et al., 2022b; Kirchmeyer et al., 2022; Kassaï Koupaï et al., 2024). These methods divide the network parameter space into domain-invariant and domain-specific parts, assuming they can represent shared and distinct knowledge in parametric PDE systems. Second, parameter conditioning schemes (Takamoto et al., 2023; Cho et al., 2024; Gupta & Brandstetter, 2022) are developed and integrated into current neural PDE solvers, allowing them to generalize across varying parameters. Third, (Hao et al., 2024a; McCabe et al., 2024; Subramanian et al., 2023) demonstrate that pretraining on diverse PDE dynamics data can enhance the transferability to downstream forecasting tasks. However, the *zero-shot OOD generalization capability* of these methods is still lacking. They demand enough test-time samples and domain-specific fine-tuning to achieve ideal performance. The core reason is that they do not explicitly illuminate the fundamental invariance principle across various PDE dynamics.

In this work, we look into the zero-shot generalizable PDE dynamics forecasting problem, with only limited variety of training trajectories available. This zero-shot setting excludes the access to test-time data for domain adaptation, which is resource-intensive and time-consuming. We are motivated by the invariant learning theory (Arjovsky et al., 2019; Liu et al., 2021), which can provably achieve ideal OOD performance by exploiting the invariant correlations between inputs and targets across varying distributions. Although invariant learning has performed impressively on vision and graph OOD tasks (Liu et al., 2022; Chen et al., 2023a; 2022), *how to define and discover the basic physical invariance principle for OOD generalizable PDE dynamics forecasting* remains unexplored. To this end, we propose to address the zero-shot OOD forecasting problem *by explicitly prescribing and estimating the PDE invariance from multiple training domains*.

To bridge this gap, we first discover that for a specific PDE system, there are two kinds of invariance independent of domain shifts: i) Individual physical processes dictated by a set of specialized operators; ii) Composition relationships between these operators and exogenous conditions like physical parameters and forcing terms. For example, reaction-diffusion systems used in chemistry and ecology (Rao et al., 2023) consist of a diffusion process formed by Laplacian operator and a nonlinear reaction function, with a diffusion and reaction coefficient controlling their rates respectively. The widely-used operator splitting method (Glowinski et al., 2017) for numerical PDE solving is built upon this discovery, which separates a complex PDE into several simpler operators and solves them by different numerical tools. Exploiting these two kinds of physics-guided invariant correlations can tackle the distribution shifts of PDE forecasting scenarios in a zero-shot manner.

In this work, informed by the two-level invariance principle in PDE systems, we propose a physics-guided invariant learning method towards zero-shot generalizable PDE dynamics forecasting. Such PDE invariance learning can be realized by an invariance-aligned network and risk equality objective. Specifically, as PDE can be split into a set of compositional operators (Glowinski et al., 2017), we design a *mixture of operator experts architecture* to capture these invariant operators and their composition relationships. It is closely aligned with the proposed two-level PDE invariance. Then, we propose a *frequency-enriched invariant learning objective* to approximate the PDE invariance by equalizing the risk of various training domains. Our main contributions are summarized as follows:

- We propose a physics-guided PDE invariance learning method termed **iMOOE**, which can achieve zero-shot PDE dynamics forecasting across diverse OOD scenarios.

- A mixture of operator expert network and a frequency-augmented risk equality objective are proposed to capture the two-fold PDE invariance.

- Extensive experiments demonstrate superior zero-shot OOD generalization capability of **iMOOE**, as well as its delicate compatibility with diverse neural operators.

## 2    OOD GENERALIZATION ON PDE FORECASTING

### 2.1    PROBLEM FORMULATION

In this work, we focus on forecasting the spatiotemporal dynamics of two-dimensional PDE systems which can be characterized in the following form:

$$\partial_t \mathbf{u} = F\left(\mathbf{x}, \mathbf{u}, \partial_\mathbf{x}\mathbf{u}, \partial_{\mathbf{xx}}\mathbf{u}, \dots, \mathbf{p}, \mathbf{f}\right), \quad \forall (t, \mathbf{x}) \in [0, T] \times \Omega, \tag{1}$$

where $\mathbf{u}(t, \mathbf{x}) \in \mathbb{R}^m$ is $m$ system state variables defined within the time span $T$ and spatial domain $\Omega \subset \mathbb{R}^2$. $\mathbf{p}$ indicates the PDE parameters that can reflect physical properties, such as the Reynold number in fluid dynamics. $\mathbf{f}$ denotes the forcing term from external input, such as the heat source in the temperature field. $F(\cdot)$ represents the *unknown* PDE law that governs the underlying physical processes. $\partial_{\mathbf{x}^n}\mathbf{u}$ is the spatial derivatives which underpin the differential operators in $F(\cdot)$. Suppose we can collect system trajectories $\{\mathbf{u}(t, \mathbf{x})\}_{t=1}^{N_t}$ of $N_t$ time steps from multiple environments $\mathcal{E}_{all}$. The environment $e \in \mathcal{E}_{all}$ can be distinguished by variable factors in PDE systems, which lead to diverse OOD scenarios. Akin to prior works on OOD dynamics forecasting (Liu et al., 2023), we consider *distribution shifts* on initial conditions $\mathbf{u}(0, \mathbf{x})$, physical parameters $\mathbf{p}$, forcing terms $\mathbf{f}$ and temporal resolution $N_t$. We also assume periodic boundary conditions for PDE systems following (Li et al., 2021; Wang et al., 2024b; Kassaï Koupaï et al., 2024). Under this multi-context setting, the goal of OOD generalizable PDE dynamics forecasting is to learn a neural simulator $f(\cdot)$ on available trajectories from limited training environments $\mathcal{D}_{tr} = \{\mathcal{D}^e\}_{e \in \mathcal{E}_{tr} \subseteq \mathcal{E}_{all}}$, and $f(\cdot)$ can perform well on all (unseen) domains *without any test-time adaptation*. This zero-shot OOD forecasting objective can be cast as a min-max risk optimization problem (Wang et al., 2022a) below:

$$\min_f \max_{e \in \mathcal{E}_{all}} \mathcal{R}^e(f), \quad \text{s.t.} \; \mathcal{R}^e(f) = \mathbb{E}_{p(\mathbf{I}^e, \mathbf{Y}^e)}\left[\ell\left(f(\mathbf{I}^e), \mathbf{Y}^e\right)\right], \tag{2}$$

where $\mathbf{I}^e = \{\mathbf{u}^e(t, \mathbf{x})\}_{t=0}^{H-1}$ is a trajectory of past $H$ steps observed from $e$, $\mathbf{Y}^e = \{\mathbf{u}^e(t, \mathbf{x})\}_{t=H}^{N_t}$ is the target sequence that should be predicted. $\ell(\cdot)$ is the loss function quantifying prediction errors. For brevity, we use $\mathbf{u}_t$ to denote $\mathbf{u}(t, \mathbf{x})$ in the rest of content. We describe the practical value of such zero-shot OOD dynamics forecasting setting in Appendix C.4.

### 2.2    INVARIANT LEARNING FOR DYNAMICS FORECASTING

Directly solving the min-max optimization problem in Eq. 2 is nontrivial. Following prior invariant learning literature (Krueger et al., 2021), we can derive the optimal solution $f^*$ by finding a maximal invariant predictor which hinges on the invariant correlations between observed trajectories $\mathbf{I}$ and future targets $\mathbf{Y}$. Let $\phi$ and $g$ denote PDE invariance extractor and output forecaster, then we can decompose $f = g \circ \phi$. In light of (Liu et al., 2021), the optimal $\phi^*(\mathbf{I})$ should satisfy two properties :

*a. Sufficiency property:* $\mathbf{Y} = g^*(\phi^*(\mathbf{I})) + \epsilon$, *where $\epsilon$ is random noise. It requires $\phi^*(\mathbf{I})$ to possess sufficient predictive information that can forecast future dynamics $\mathbf{Y}$.*

*b. Invariance property:* $\mathbb{E}_{p^e}\left[\ell\left(g^*(\phi^*(\mathbf{I}^e)), \mathbf{Y}^e\right)\right] = \mathbb{E}_{p^{e'}}[\ell(g^*(\phi^*(\mathbf{I}^{e'})), \mathbf{Y}^{e'})], \forall e, e' \in \mathcal{E}_{all}$. *It requires $\phi^*$ to identify the invariance principle in PDE dynamical system $F(\cdot)$. Such PDE invariance can give rise to equal risks across different forecasting environments.*

Based on above two requisites for PDE invariance learning, the core challenge lies in how to identify the fundamental PDE invariance principle from multi-domain trajectories. The forecasting model $f^* = g^*(\phi^*(\mathbf{I}))$ built upon PDE invariance can realize the desirable zero-shot OOD performance.

### 2.3    INVARIANCE PRINCIPLE FOR PDE DYNAMICS

We now formally introduce the invariance principle for PDE dynamics forecasting. Due to the lack of such invariance formalism, domain-invariant representations learned by previous meta-learning-based (Kirchmeyer et al., 2022; Kassaï Koupaï et al., 2024) or parameter conditioning-based methods (Takamoto et al., 2023) can not achieve ideal OOD outcomes. We define PDE invariance based on such finding: As the PDE law $F(\cdot)$ is composed of a few operator items (Rudy et al., 2017), the widely used operator splitting method can (Glowinski et al., 2017) decompose PDE into different operators and combine the solution of each part. It has exhibited great efficiency on a wide range

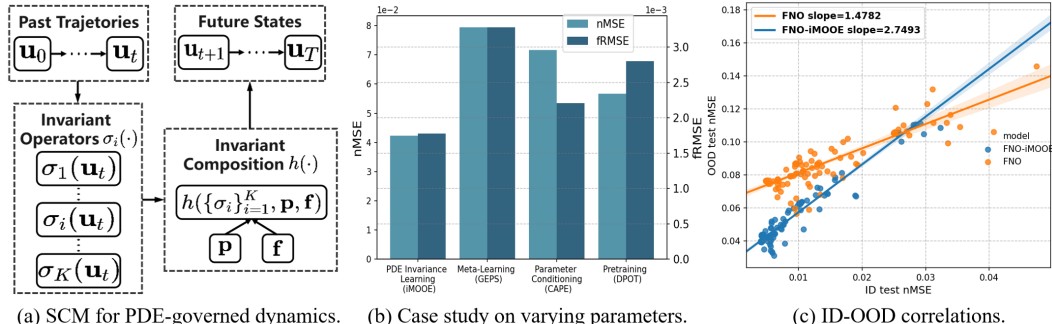

(a) SCM for PDE-governed dynamics.     (b) Case study on varying parameters.     (c) ID-OOD correlations.

Figure 1: (a) The SCM diagram for the formation process PDE dynamics. It illustrates prescribed two-level PDE invariance and potential distribution shifts on exogenous inputs. (b) A case study by varying physical parameters of DR dynamics to compare the zero-shot OOD performance of four methods. Without the guidance of formalized PDE invariance, previous methods can not achieve better OOD results on unseen environments. (c) The ID-OOD correlations of two neural operators. Based on the slope of two linear positive ID-OOD lines, FNO equipped with PDE invariance learning can capture more transferrable knowledge from limited training domains and achieve better OOD robustness. Refer to Appendix F.6 for more details.

of PDE solving, including the complex nonlinear Navier-Stokes equation (Glowinski et al., 2017). Then, we derive the two-fold PDE invariance principle which underpins PDE system evolution:

*(i). Operator invariance: PDE dynamics are governed by the composition of a few spatial operators $\{\sigma_i(\mathbf{x}, \mathbf{u}, \partial_{\mathbf{x}}\mathbf{u}, \dots)\}_{i=1}^{K}$. These elementary operators representing distinct physics remain invariant across system evolution and different domains.*

*(ii). Compositionality invariance: The composition method $h$ to aggregate basic operators, physical parameters and forcing terms is fixed as $F = h(\sigma_1, ..., \sigma_i, ..., \sigma_K, \mathbf{p}, \mathbf{f})$ for a specific PDE system.*

In addition, the future state $\hat{\mathbf{u}}_{t+1}$ can be calculated as $\hat{\mathbf{u}}_{t+1} = \int_{t}^{t+1} h(\{\sigma_i\}_{i=1}^{K}, \mathbf{p}, \mathbf{f})\mathrm{d}t + \mathbf{u}_t$ given the last observation $\mathbf{u}_t$. In a nutshell, invariant correlations between input $\mathbf{I}$ and target $\mathbf{Y}$ involve: invariant operators $\{\sigma_i\}_{i=1}^{K}$, invariant compositional relationships $h$ among different items, as well as the fixed step-wise numerical integration. We present a structural causal model (SCM) (Krueger et al., 2021) in Fig. 1(a) to illustrate the formation process of PDE dynamics and its pertinent two-level invariant correlations. Regardless of various time steps and distribution shifts on $\{\mathbf{u}_0, \mathbf{p}, \mathbf{f}\}$, the fundamental set of operators and their composition relationships can remain invariant. We formally justify the proposed two-level invariance principle can hold across many specialized PDE systems from three perspectives in Appendix B. Besides, we provide a case study by varying diffusion and reaction coefficient of DR dynamics in Fig. 1(b), and the ID-OOD correlation lines (an effective metric to assess OOD robustness (Yuan et al., 2023)) in Fig. 1(c). Both of them can further demonstrate the effectiveness of the proposed physics-guided PDE invariance learning for improving zero-shot OOD capability. Refer to Appendix C for more related works on PDE dynamics forecasting and invariant learning.

## 3   PHYSICS-GUIDED INVARIANT LEARNING FRAMEWORK

To develop the physics-guided invariant learning for zero-shot OOD forecasting, the key challenge resides in how to cultivate an effective invariant forecaster that can exploit two-level PDE invariance principle defined in Sec. 2.3. To achieve this, we first design a mixture of operator experts network which can respect the invariant correlations between past observations and future trajectories. In vision OOD tasks, the mixture-of-experts (MoE) architecture has shown great generalization, since MoE can closely align with the invariant correlations between image attributes and labels (Li et al., 2023a). But how to enable MoE to capture PDE invariance for zero-shot OOD forecasting remains an open issue. Next, we propose a frequency-enriched invariant learning objective to estimate PDE invariance from multiple training domains. It can tackle the high-frequency learning pitfall in existing neural operators (Khodakarami et al., 2025). Up to now, we can derive the invariant Mixture

of Operator Experts (iMOOE), a physics-guided invariant learning method towards zero-shot OOD generalizable PDE dynamics forecasting as depicted in Fig. 2.

## 3.1 Architecture Alignment: Mixture of Operator Experts

To align with the operator and compositionality invariance presented in Sec. 2.3, we develop the MOOE architecture which consists of two parts: i) A group of specialized neural operator experts to represent the unique and unknown physics. ii) A fusion network to aggregate these expert output with exogenous input like system parameters. This design shares the similar spirit with the effective operator splitting solver (Glowinski et al., 2017), which separates a complex PDE into a set of simpler operators and calculates each part by suitable numerical methods. Taking the reaction-diffusion equation as an example (Krishnapriyan et al., 2021), we can solve the second-order diffusion component by finite difference and calculate the reaction function by forward pass. Note that the typical operator splitting algorithm for PDE solving has a serial structure, which treats the solution of the former operator as the initial condition of the latter operator. But such serial operator solving will lead to slow computation for neural PDE learning. In this regard, the developed MOOE network stacks operator experts in parallel instead of linking them in series, as depicted in Fig. 2(a).

**Operator Experts.** Similar to operator splitting, each individual operator expert in MOOE should be specialized in approximating a distinct physical process. In addition, we observe that each operator component in PDEs is formed by state variable $\mathbf{u}$ or its certain orders of partial derivatives. For instance, advection term $\nabla \cdot \mathbf{u}$ consists of $\partial_{\mathbf{x}}\mathbf{u}$ whereas diffusion term $\nabla^2 \mathbf{u}$ stems from $\partial_{\mathbf{xx}}\mathbf{u}$. In this regard, we design a binary mask vector $\mathbf{m}_i = \{0,1\}^S$ for each operator expert $\sigma_i$, making it can adaptively select the useful spatial derivatives to benefit operator learning. Here, $S$ is the number of pre-computed derivatives $\partial_{\mathbf{x}^n}\mathbf{u}$. We leverage existing neural operators (Kovachki et al., 2023) as the backbone operator experts, which excel at approximating PDE laws:

$$\sigma_i = \text{NO}_i\left(\mathbf{x}, \mathbf{u}_{t-W+1:t}, \mathbf{m}_i \odot [\partial_{\mathbf{x}}\mathbf{u}_t, \partial_{\mathbf{xx}}\mathbf{u}_t, \dots]^{\mathbb{T}}\right). \quad (3)$$

A notable advantage is that expert $\text{NO}_i(\cdot)$ can be compatible with a broad variety of neural operators *without any modification* on their structures. We verify this compatibility in Section 4.3. Apart from spatial coordinates $\mathbf{x}$ and past sequences of length $W$, we also incorporate pre-calculated derivatives as prior input, which can render operator learning easier (Li et al., 2024a). To encourage operator experts to represent inhomogeneous physical processes, we feed them with different sets of pre-calculated derivatives by designing a mask diversity loss:

$$\mathcal{L}_{mask} = \min_{\{\mathbf{m}_i\}_{i=1}^K} \frac{1}{K^2} \sum_{i=1}^K \sum_{j=1}^K \exp\left(-\|\mathbf{m}_i - \mathbf{m}_j\|_2^2\right). \quad (4)$$

We illustrate this masking-based input derivative selection design in Fig. 2(b), and analyze its effect in Appendix F.1.2.

**Fusion Network.** The key role of the fusion network is to aggregate the output of operator experts and condition it on physical parameters in variable $\mathbf{p}$ and $\mathbf{f}$. Regarding the parameter conditioning, we directly concatenate the expert output with PDE parameters and utilize a multi-layer perceptron (MLP) to encode it. As for the aggregator, we should consider two different cases after some empirical trials: i) For PDE systems with strong non-linearity, such as the convection term $\mathbf{u} \cdot \nabla(\nabla \times \mathbf{u})$ in turbulence flow (Dresdner et al., 2023), we employ an extra network to learn this complex composition. ii) For PDE systems without these intractable non-linear terms, such as the additive operator relationship in reaction-diffusion, we can simply add up expert output. In brief, the fusion network representing the invariant composition relationship $h$ can be expressed as follows:

$$h = \text{FusionNet}\left(\text{MLP}_1\left(\sigma_i, \mathbf{p}, \mathbf{f}\right), \dots, \text{MLP}_K\left(\sigma_K, \mathbf{p}, \mathbf{f}\right)\right). \quad (5)$$

In Appendix F.1.3, we verify that choosing a suitable type of fusion network can better align with the compositionality invariance and achieve better OOD performance.

## 3.2 Frequency-Enriched Invariant Learning Objective

With the MOOE network which can align with two-fold PDE invariance, the next step is to design an effective invariant learning objective that can satisfy two requisites presented in Sec. 2.2. However,

(a) Overview of iMOOE framework.      (b) Individual operator experts.

Figure 2: (a) Overview of iMOOE method, which can capture the physics-guided PDE invariance by the mixture of operator experts architecture and frequency-enriched multi-context ($|\mathcal{E}_{tr}| = 2$ here) training. (b) The structure of single operator expert, which can well fit in diverse neural operators.

we find that the intrinsic spectral bias issue of neural operators will hinder PDE invariance learning due to the neglect of high-frequency information. To mitigate it, we propose a frequency-augmented invariant learning loss which can help to capture the complete domain-generalizable representations in PDE dynamics.

**Maximal Prediction Loss.** Following invariant learning literature (Liu et al., 2021), we can fulfill the sufficiency property by maximizing the mutual information between the two-level PDE invariance $h \circ \{\sigma_i\}_{i=1}^K$ and future forecasts $\mathbf{Y}^e$. In light of (Tsai et al., 2021), this information maximization objective can be realized by the maximal prediction loss in dynamics forecasting:

$$\mathcal{L}_{pred} = \frac{1}{|\mathcal{E}_{tr}|} \sum_{e \in \mathcal{E}_{tr}} \mathcal{R}_{pred}^e, \text{ and } \mathcal{R}_{pred}^e = \mathbb{E}_{p^e} \left[ \sum_{t=H}^{N_t} \left\| \mathbf{u}_t^e - \int_{t-1}^{t} h(\{\sigma_i\}_{i=1}^K, \mathbf{p}^e, \mathbf{f}^e) \mathrm{d}t - \hat{\mathbf{u}}_{t-1}^e \right\|_2^2 \right],$$
(6)

where we utilize the autoregressive training manner and $\hat{\mathbf{u}}_{t-1}^e$ is the predicted state at the last time step. We utilize the Euler forward method to implement the numerical integration on time marching.

**Risk Equality Loss.** The invariance property demands prediction error across various environments to be equal. As proved in invariant learning literature (Krueger et al., 2021), we can meet this risk equality objective by minimizing the variance of risks over different training environments:

$$\mathcal{L}_{inv} = \mathrm{Var} \left( \{\mathcal{R}_{pred}^e\}_{e \in \mathcal{E}_{tr}} \right),$$
(7)

where $\mathcal{R}_{pred}^e$ is provided in Eq. 6. We borrow the useful linear scheduling scheme to impose this risk equality loss (Krueger et al., 2021), which reserves an initial empirical risk minimization stage (i.e. a pretraining stage merely by $\mathcal{L}_{pred}$) to learn the rich predictive representations. Refer to Appendix F.1.5 for the effect of this linear invariant loss scheduling. Note that our design differs from (Krueger et al., 2021) in environment division. In addition to taking physical parameters in $\mathbf{p}$ and $\mathbf{f}$ as environment labels, we also partition training domains by different autoregressive steps, since there exist covariate shifts in $p(\mathbf{I}^e, \mathbf{Y}^e)$. Specifically, during the autoregressive prediction, the distribution of past sequences $p(\mathbf{I}^e)$ can change with the time marching, but the correlations between $\mathbf{I}^e$ and $\mathbf{Y}^e$ keep invariant at each time step. We find this step-wise division is instrumental for fluid dynamics forecasting such as Navier-Stokes and Burgers systems, as shown in Appendix F.1.4.

**Frequency Enrichment Loss.** Both $\mathcal{L}_{pred}$ and $\mathcal{L}_{inv}$ are inadequate to capture the complete PDE invariance, since neural operators prioritize learning the dominant low-frequency features in state $\mathbf{u}$ (a.k.a. the spectral bias issue) (Lippe et al., 2023). Ignoring the necessary high-frequency modes entails spectral information loss for invariant operator learning, which impedes $\sigma_i$ to satisfy the sufficiency property given in Sec. 2.2. Besides, high-frequency learning errors can propagate to the whole spectral domain during the autoregressive prediction process, rendering it hard to generalize across OOD scenarios with different frequency distributions. To this end, we propose to augment high-frequency representations when learning PDE invariance by designing a regularization item:

$$\mathcal{L}_{freq} = \frac{1}{|\mathcal{E}_{tr}|} \sum_{e \in \mathcal{E}_{tr}} \mathcal{R}_{freq}^e, \text{ and } \mathcal{R}_{freq}^e = \mathbb{E}_{p^e} \left[ \sum_{t=H}^{N_t} \sum_{\xi} \|\xi\|_2^2 \|\mathcal{F}(\mathbf{u}_t)(\xi) - \mathcal{F}(\hat{\mathbf{u}}_t)(\xi)\|_2^2 \right],$$
(8)

where $\mathcal{F}$ is the fast Fourier transform and $\xi$ is the wavenumber vector for each spatial frequency. Apparently, the weight $\|\xi\|_2^2$ can pay more attention to the high-frequency modes at each forecasting

step. We validate such frequency enrichment loss can induce better PDE forecasting generalization in Appendix. F.1.1. Prior works on OOD vision recognition (Chen et al., 2023a; Zhang et al., 2022) also proved that diverse and rich features can lead to better OOD capability.

### 3.3 Overall Framework

The total PDE invariance learning objective for iMOOE is presented below:

$$\mathcal{L}_{total} = \lambda_{pred}\mathcal{L}_{pred} + \lambda_{inv}\mathcal{L}_{inv} + \lambda_{freq}\mathcal{L}_{freq} + \lambda_{mask}\mathcal{L}_{mask}; \tag{9}$$

Equipped with this hybrid training loss and invariance-aligned architecture developed in Sec. 3.1, we can effectively learn the proposed PDE invariance to achieve zero-shot OOD forecasting. Existing neural operators always train with prediction loss $\mathcal{L}_{pred}$, without any effort to learn the fundamental PDE invariance principle. This could be the key reason for their failures on OOD dynamics forecasting. We demonstrate in Section 4.3 that when equipped with the explicit physics-informed PDE invariance learning method iMOOE, current neural operators can realize better OOD performance. Moreover, in Appendix F.7, we further investigate how the properties of multi-environment training data can affect the zero-shot OOD capability of iMOOE. As simulating PDE trajectories or measuring real-world PDE dynamics is expensive, such analysis can provide a guideline on how to collect training data under a limited data budget.

## 4 Experiments

### 4.1 Experimental Setup

**Datasets.** We adopt five PDE dynamical systems in different fields for the spatiotemporal physical dynamics forecasting task: Diffusion-Reaction (DR) (Takamoto et al., 2022), Navier-Stokes (NS) (Li et al., 2021), Burgers (BG) (Hao et al., 2024b), Shallow-Water (Takamoto et al., 2022) and Heat-Conduction (HC) (Hao et al., 2024b). We construct a wide range of OOD scenarios by varying the physical parameters of initial conditions $\mathbf{u}_0$, PDE coefficients $\mathbf{p}$, forcing terms $\mathbf{f}$ or temporal resolutions $N_t$. ID and OOD parameters for PDE simulation are randomly drawn from two non-overlapped uniform distributions, while previous parametric PDE learning works like (Kassaï Koupaï et al., 2024; Takamoto et al., 2023) just select several separate parameters. The spatial resolution of each state frame is fixed to $64 \times 64$. See Appendix D for detailed description on data generation.

**Evaluation Criteria.** We leverage two metrics in PDEBench (Takamoto et al., 2022) to comprehensively evaluate the forecasting performance: i) normalized Mean Squared Error (nMSE) in raw data space: $\text{nMSE} = \|\hat{\mathbf{u}}_{H:N_t} - \mathbf{u}_{H:N_t}\|_2^2 / \|\mathbf{u}_{H:N_t}\|_2^2$. ii) fourier Root Mean Squared Error (fRMSE) in frequency domain: $\text{fRMSE} = \sqrt{\sum_{\xi_{\min}}^{\xi_{\max}} \|\mathcal{F}(\hat{\mathbf{u}}_{H:N_t})(\xi) - \mathcal{F}(\mathbf{u}_{H:N_t})(\xi)\|_2^2 / (\xi_{\max} - \xi_{\min} + 1)}$. They can reflect the forecasting accuracy of PDE system states from both data and physics views. Note that for all experiments, we present both *in-distribution* (ID) and *out-of-distribution* (OOD) results in *zero-shot setting* (i.e. without any access to test-time samples for adaptation).

**Implementation Details.** We fix the number of operator experts $K = 2$ and loss weights $\lambda_{pred} = 1, \lambda_{freq} = 0.1, \lambda_{mask} = 0.001$. Similar to prior invariant learning work (Krueger et al., 2021), we linearly schedule $\lambda_{inv}$ with an upper threshold of $0.001$. The popular Fourier Neural Operator (FNO) (Li et al., 2021) with 4 layers and 64 width is employed as the backbone of each operator expert. We pre-calculate first- and second-order spatial derivatives for adaptive selection by masking. Past $H = W = 10$ steps observations are used to predict future trajectories, following the same setting in previous PDE forecasing works (Li et al., 2021; Kassaï Koupaï et al., 2024). iMOOE is trained on a NVIDIA A100 GPU with total 500 epochs, $0.001$ initial learning rate by Adam optimizer.

### 4.2 Zero-shot OOD Performance

**Baselines.** We select six latest PDE forecasting methods with highlighted OOD generalization capability: i) CoDA (Kirchmeyer et al., 2022) and GEPS (Kassaï Koupaï et al., 2024): two context-aware meta-learning-based models. ii) CAPE (Takamoto et al., 2023): a parameter conditioning method. iii) CNO (Raonic et al., 2023): a robust convolutional neural operator. iv) DPOT (Hao et al., 2024a): a transformer-based operator with denoising pretraining. v) VCNeF (Hagnberger et al., 2024): a

Table 1: Zero-shot ID/OOD generalization results compared to existing generalizable PDE dynamics forecasting methods. The listed five PDE dynamical systems are employed to synthesize a diversity of OOD forecasting scenarios. Best results are in **bold** and second-best results are underlined. "n.a." indicates the excess of computational resource limit.

| Metrics | Models | DR | | NS | | BG | | SW | | HC | |
|---|---|---|---|---|---|---|---|---|---|---|---|
| | | ID | OOD | ID | OOD | ID | OOD | ID | OOD | ID | OOD |
| nMSE | CoDA | 3.40e-1 | 6.05e-1 | 4.31e-1 | 9.14e-1 | 8.72e-1 | 9.22e-1 | n.a. | n.a. | 1.16e+0 | 2.37e+0 |
| | CAPE | 8.90e-3 | 7.16e-2 | 9.09e-2 | 3.56e-1 | 5.00e-3 | 3.04e-2 | 2.71e-6 | 6.18e-5 | 5.70e-2 | 3.65e+0 |
| | CNO | 3.36e+0 | 2.56e+0 | 6.03e-1 | 6.90e-1 | 3.30e-3 | 1.87e-2 | 2.10e-5 | 3.82e-5 | 1.01e-1 | 2.45e+0 |
| | DPOT | 2.62e-2 | 5.67e-2 | 3.44e-1 | 5.08e-1 | 2.18e-2 | 8.41e-2 | 6.69e-5 | 4.85e-4 | **3.68e-2** | 2.12e+0 |
| | VCNeF | 8.70e-3 | 7.84e-2 | 1.40e-1 | 3.81e-1 | 1.03e-2 | 4.68e-2 | 4.59e-5 | 6.12e-4 | 1.37e+0 | 1.42e+0 |
| | GEPS | 8.71e-3 | 7.94e-2 | 2.07e-1 | 4.13e-1 | 2.24e-2 | 7.56e-2 | 1.22e-4 | 2.76e-4 | 9.43e-1 | 1.35e+0 |
| | **iMOOE** | **5.15e-3** | **4.23e-2** | **6.49e-2** | **3.12e-1** | **1.20e-3** | **1.08e-2** | **3.34e-7** | **3.02e-5** | 3.92e-2 | **1.22e+0** |
| fRMSE | CoDA | 7.88e-3 | 9.93e-3 | 3.81e-2 | 7.31e-2 | 2.12e-2 | 2.50e-2 | n.a. | n.a. | 1.25e-2 | 7.09e-3 |
| | CAPE | 1.18e-3 | 2.21e-3 | 1.97e-2 | 5.77e-2 | 2.13e-3 | 5.70e-3 | 1.22e-4 | 5.50e-4 | 2.05e-3 | 8.83e-3 |
| | CNO | 3.06e-2 | 2.43e-2 | 4.67e-2 | 7.79e-2 | 2.60e-3 | 5.82e-3 | 3.35e-4 | 4.79e-4 | 2.84e-3 | 7.54e-3 |
| | DPOT | 3.00e-3 | 2.80e-3 | 4.59e-2 | 7.30e-2 | 5.32e-3 | 1.04e-2 | 6.61e-4 | 1.95e-3 | 3.08e-3 | 7.83e-3 |
| | VCNeF | 1.68e-3 | 2.77e-3 | 2.66e-2 | 6.11e-2 | 3.20e-3 | 7.28e-3 | 5.46e-4 | 1.93e-3 | 1.26e-2 | 7.13e-3 |
| | GEPS | 1.99e-3 | 3.28e-3 | 3.85e-2 | 6.85e-2 | 5.14e-3 | 9.38e-3 | 9.08e-4 | 1.38e-3 | 1.04e-2 | 6.16e-3 |
| | **iMOOE** | **9.16e-4** | **1.78e-3** | **1.38e-2** | **5.36e-2** | **1.10e-3** | **3.83e-3** | **4.59e-5** | **3.65e-4** | **1.31e-3** | **5.95e-3** |

Table 2: Zero-shot time extrapolation results on two PDE systems.

| Models | DR | | | | NS | | | |
|---|---|---|---|---|---|---|---|---|
| | nMSE | | fRMSE | | nMSE | | fRMSE | |
| | In-time | Out-time | In-time | Out-time | In-time | Out-time | In-time | Out-time |
| CAPE | 3.30e-3 | 3.88e-1 | 1.11e-3 | 8.58e-3 | 1.92e-1 | 5.46e-1 | 4.36e-2 | 7.04e-2 |
| DPOT | 4.51e-3 | 4.58e+0 | 1.31e-3 | 1.75e-2 | 1.94e-1 | 6.22e-1 | 4.62e-2 | 7.88e-2 |
| VCNeF | 2.41e-3 | 5.46e-1 | 9.81e-4 | 1.26e-2 | 3.00e-1 | 9.67e-1 | 4.90e-2 | 8.66e-2 |
| GEPS | 2.52e-3 | 6.96e-1 | 1.02e-3 | 1.39e-2 | 2.70e-1 | 6.57e-1 | 4.78e-2 | 7.77e-2 |
| **iMOOE** | **9.47e-4** | **1.99e-1** | **4.93e-4** | **6.26e-3** | **1.65e-1** | **4.57e-1** | **3.89e-2** | **6.79e-2** |

conditional neural field-based method. Note that meta-learning-based methods commonly require few-shot adaptation to perform OOD forecasting. In Appendix F.4, we describe how to adapt them to zero-shot setting and further compare zero-shot iMOOE with few-shot CoDA, GEPS. Implementation details of these baseline models are provided in Appendix F.9.

**Results.** We report ID/OOD generalization outcomes on various unseen scenarios in Table 1. It is obvious that iMOOE can achieve the state-of-art (SOTA) results on this simulated benchmark, with an average increase of 40.21% on nMSE and 30.78% on fRMSE. Such considerable promotion reflects that explicitly learning the proposed physics-guided PDE invariance can boost zero-shot OOD performance on PDE dynamics forecasting. Moreover, we present the OOD results on extrapolated temporal resolutions in Table 2. Following previous time extrapolation setting (Kassaï Koupaï et al., 2024), we train on $[0, N_t]$ and test on $[0, 2N_t]$. We find that iMOOE can achieve SOTA results with an average growth of 32.51% on nMSE and 15.30% on fRMSE. It indicates that learning the underlying PDE invariance across time steps can improve the OOD performance on unseen temporal distribution shift scenarios. To measure iMOOE's zero-shot OOD capacity more clearly, we present an empirical upper bound for its OOD performance in Appendix F.5.

## 4.3 UNIVERSALITY STUDY

In Table 3, we manifest iMOOE's flexibility on integrating diverse operator learning models into operator experts $\sigma(\cdot)$ in a plug-and-play fashion. We involve four classic categories of neural operators including FNO (Li et al., 2021), DeepONet (Lu et al., 2021), neural field-based VCNeF (Hagnberger et al., 2024) and transformer-based OFormer (Li et al., 2023b). We validate their vanilla capability and iMOOE-upgraded performance on DR dynamics under 8 OOD environments, and present OOD nMSE results of each environment in Table 3. Existing neural operators have not been comprehensively validated under this zero-shot OOD setting. When augmented by either MOOE or iMOOE, these neural operators can consistently achieve lower mean and variance values on nMSE over various OOD contexts. Such promotion underscores both the PDE invariance-aligned architecture and frequency-enriched objective can improve zero-shot OOD capability of existing neural operators.

Table 3: Operator compatibility study on DR data with various OOD contexts. "Env1" to "Env8" indicates eight different settings for diffusion and reaction coefficients. "+MOOE" denotes employing vanilla neural operators as the backbone of operator experts. "+iMOOE" denotes further imposing the frequency-enriched invariance training on MOOE.

| Operators | Variants | Env1 | Env2 | Env3 | Env4 | Env5 | Env6 | Env7 | Env8 | Mean | Std |
|---|---|---|---|---|---|---|---|---|---|---|---|
| FNO | Naive | 6.78e-2 | 8.80e-2 | 6.70e-2 | 6.00e-2 | 3.14e-2 | 1.11e-1 | 1.62e-1 | 4.68e-2 | 7.94e-2 | 3.88e-2 |
| | +MOOE | 3.40e-2 | 6.00e-2 | 3.88e-2 | 3.28e-2 | 1.88e-2 | 8.14e-2 | 1.22e-1 | 2.49e-2 | 5.16e-2 | 3.26e-2 |
| | +iMOOE | 3.19e-2 | 5.20e-2 | 3.12e-2 | 3.05e-2 | 1.41e-2 | 6.02e-2 | 9.93e-2 | 1.87e-2 | **4.23e-2** | **2.60e-2** |
| DeepONet | Naive | 5.94e-1 | 9.73e-1 | 6.38e-1 | 5.18e-1 | 3.99e-1 | 7.27e-1 | 5.58e-1 | 5.09e-1 | 6.15e-1 | 1.63e-1 |
| | +MOOE | 6.03e-1 | 9.74e-1 | 6.39e-1 | 5.37e-1 | 3.70e-1 | 7.18e-1 | 5.56e-1 | 4.82e-1 | 6.10e-1 | 1.69e-1 |
| | +iMOOE | 5.45e-1 | 8.75e-1 | 5.76e-1 | 4.93e-1 | 3.47e-1 | 6.31e-1 | 4.82e-1 | 4.45e-1 | **5.49e-1** | **1.47e-1** |
| VCNeF | Naive | 5.77e-2 | 9.74e-2 | 6.11e-2 | 5.35e-2 | 2.48e-2 | 1.26e-1 | 1.68e-1 | 3.85e-2 | 7.84e-2 | 4.54e-2 |
| | +MOOE | 3.51e-2 | 6.91e-2 | 3.37e-2 | 3.31e-2 | 2.62e-2 | 8.86e-2 | 1.40e-1 | 3.31e-2 | 5.73e-2 | 3.72e-2 |
| | +iMOOE | 3.29e-2 | 6.59e-2 | 3.01e-2 | 3.30e-2 | 2.60e-2 | 8.31e-2 | 1.37e-1 | 3.36e-2 | **5.52e-2** | **3.62e-2** |
| OFormer | Naive | 4.95e-2 | 6.74e-2 | 4.75e-2 | 4.74e-2 | 5.27e-2 | 6.77e-2 | 7.47e-2 | 5.31e-2 | 5.75e-2 | 1.01e-2 |
| | +MOOE | 4.47e-2 | 4.38e-2 | 4.16e-2 | 4.98e-2 | 5.31e-2 | 4.78e-2 | 6.60e-2 | 5.03e-2 | 4.96e-2 | 7.12e-3 |
| | +iMOOE | 4.06e-2 | 4.15e-2 | 4.17e-2 | 4.80e-2 | 5.29e-2 | 3.47e-2 | 3.65e-2 | 5.09e-2 | **4.34e-2** | **6.18e-3** |

Table 4: Zero-shot OOD results on SST dynamics.

| Models | nMSE | | fRMSE | |
|---|---|---|---|---|
| | Mean | Std | Mean | Std |
| GEPS | 8.24e-1 | 3.08e-1 | 5.28e-2 | 6.02e-3 |
| DPOT | 5.56e-1 | 2.43e-1 | 3.65e-2 | 5.19e-3 |
| VCNeF | 6.69e-1 | 2.67e-1 | 4.61e-2 | 6.39e-3 |
| DyAd | 5.87e-1 | 2.44e-1 | 3.79e-2 | 5.15e-3 |
| CAPE | 6.51e-1 | 2.81e-1 | 3.84e-2 | 5.59e-3 |
| iMOOE | **5.12e-1** | **2.36e-1** | **3.44e-2** | **5.03e-3** |

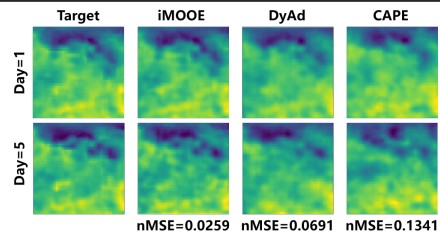

Figure 3: Test SST sample showcase.

## 4.4 APPLICATION TO REAL-WORLD PDE DYNAMICS

Apart from the simulated benchmark above, we also leverage two real-world PDE-governed Ocean-Atmosphere dynamics datasets, including Sea Surface Temperature (SST) (Huang et al., 2021) and stereo Sea Surface Elevation (Guimarães et al., 2020) to validate iMOOE's generalization capability. These two datasets represent upper-ocean thermodynamics and free-surface ocean wave dynamics respectively, and contain sensory measurement noise. In SST forecasting setting, a specific region on Pacific Ocean is selected and divided into $60 \times 60$ grid. We predict SST state of future 6 days using past 4 days observations. SST data between year 1982-2019 and 2020-2021 is utilized for training and testing. Note that each independent SST trajectory can be deemed as an instance from a unique environment, since daily SST variations are affected by many meteorological conditions like solar radiation and wind speed. Input parameters are also unknown so we feed one-valued vector to the fusion network. In SSE forecasting setting, we choose the wave dynamics recorded at La Jument lighthouse with total 4500 frames on $241 \times 221$ grid. The training and testing sets are formed by the first 4000 frames and remaining 500 frames. We input past 4 steps SSE state to predict future 6 steps. We take a typical ocean dynamics forecasting baseline called DyAd (Wang et al., 2022b) and present OOD comparison results in Table 4, 5. We find that iMOOE can attain the lowest mean and variance on two metrics across various OOD samples, which reflects iMOOE's capability to capture the underlying physics law in real-world ocean dynamics. Test samples in Fig. 3, 4 exhibit iMOOE can capture local variations of SST and SSE with higher fidelity and accuracy. Beyond 2D PDE-governed dynamics, we further demonstrate iMOOE's zero-shot OOD capability can be extended to other types of dynamical systems in Appendix F.2.

Table 6: Effect of varying numbers of operator experts.

| Number of expert $K$ | DR | | BG | | Inference time |
|---|---|---|---|---|---|
| | nMSE | fRMSE | nMSE | fRMSE | |
| 1 | 5.26e-2 | 1.94e-3 | 1.38e-2 | 4.35e-3 | 0.08s |
| 2 | 4.63e-2 | 1.81e-3 | 1.14e-2 | 3.95e-3 | 0.11s |
| 3 | **4.17e-2** | **1.74e-3** | **1.07e-2** | **3.83e-3** | 0.15s |
| 4 | 5.00e-2 | 1.84e-3 | 1.14e-2 | 3.88e-3 | 0.18s |

Table 5: Zero-shot OOD results on SSE dynamics.

| Models | nMSE | | fRMSE | |
|---|---|---|---|---|
| | Mean | Std | Mean | Std |
| GEPS | 3.41e-2 | 4.96e-3 | 4.11e-3 | 1.55e-4 |
| DPOT | 2.46e-2 | 3.28e-3 | 3.54e-3 | 1.49e-4 |
| VCNeF | 3.41e-2 | 4.96e-3 | 4.11e-3 | 1.55e-4 |
| DyAd | 3.17e-2 | 4.61e-3 | 3.96e-3 | 1.50e-4 |
| CAPE | 3.34e-2 | 4.85e-3 | 4.07e-3 | 1.53e-4 |
| **iMOOE** | **1.52e-2** | **2.34e-3** | **2.78e-3** | **1.42e-4** |



Figure 4: Test SSE sample showcase.

## 4.5 SENSITIVITY ANALYSIS

In Table 6, we investigate the influence of the number of operator experts $K$ on iMOOE's zero-shot OOD performance. To ensure a fair comparison, we only escalate $K$ from 1 to 4 and keep other setups like the width of FNO and training batch size unchanged. We find that the best-performing group is $K = 3$ while the worst setting is $K = 1$. This reveals that small $K$ (i.e. only 1 expert) is not sufficient to capture the operator invariance, while large $K$ (i.e. 4 experts) could be redundant given that actual PDE systems contains only a few number of compositional invariant operators (Rudy et al., 2017). Besides, the increasing number of neural operators can exacerbate the computational overhead. Refer to Appendix E for more detailed explanations on the effect of expert number $K$ and its distinction from the mixture-of-expert (MoE) architecture in large foundation models (LFMs). Refer to Appendix F.3 for more sensitivity analysis on loss weights in Eq. 9.

## 5 CONCLUSION

In this work, we propose the iMOOE learning framework to address the zero-shot OOD generalization issue in the scope of PDE-governed spatiotemporal physical dynamics forecasting. We first introduce the two-level physics-guided invariance principle for PDE dynamical systems. Then, we develop the mixture of operator experts architecture plus the frequency-augmented invariant learning objective to capture such PDE invariance from limited training environments. Various experiments demonstrate the excellent zero-shot OOD forecasting capability of iMOOE. However, the proposed PDE invariance learning is validated on a limited diversity of dynamical systems. In future work, we plan to extend iMOOE's zero-shot OOD capability to other types of PDE dynamics, such as PDE systems on irregular grids, or more real-world applications like earth system forecasting.

### ETHICS STATEMENT

Our work is only aimed at generalizable PDE dynamics forecasting for human good, so there is no involvement of human subjects or conflict of interests as far as the authors are aware of.

### ACKNOWLEDGEMENTS

This work was supported in part by the National Key R&D Program of China (Grant No.2023YFF0725001), in part by the National Natural Science Foundation of China (Grant No.92370204), in part by the guangdong Basic and Applied Basic Research Foundation (Grant No.2023B1515120057), in part by the Key-Area Special Project of Guangdong Provincial Ordinary Universities(2024ZDZX1007).

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

# A  LLM USAGE

Large Language Models (LLMs) were used to aid in the writing and polishing of the manuscript. Specifically, we used an LLM to assist in refining the language, improving readability, and ensuring clarity in various sections of the paper. The model helped with tasks such as sentence rephrasing, grammar checking, and enhancing the overall flow of the text. It is important to note that the LLM was not involved in the ideation, research methodology, or experimental design. All research concepts, ideas, and analyses were developed and conducted by the authors. The contributions of the LLM were solely focused on improving the linguistic quality of the paper, with no involvement in the scientific content or data analysis. We have ensured that the LLM-generated text adheres to ethical guidelines and does not contribute to plagiarism or scientific misconduct.

# B  RIGOROUS JUSTIFICATION FOR TWO-LEVEL PDE INVARIANCE

Firstly, in the field of PDE learning, the commonly utilized assumption is that many specialized PDE systems demonstrate compositional structures and contains a group of distinct physical processes, as claimed in Section 4.1 of (McCabe et al., 2024). For example, Navier-Stokes systems are composed of convection, pressure and viscosity, wave systems are composed of Hamiltonian flow and dissipation. This PDE compositionality assumption can be formulated as (Xu et al., 2025): $\partial_t \mathbf{u}(\mathbf{x}) = F(\sigma_1(\mathbf{u}, \mathbf{x}), ..., \sigma_i(\mathbf{u}, \mathbf{x}), ..., \sigma_K(\mathbf{u}, \mathbf{x}))$. $F(\cdot)$ is a nonlinear function, which represents the compositional relationship among a group of basic operators $\{\sigma_i(\mathbf{u}, \mathbf{x})\}_{i=1}^K$. Such compositional operator structure guides us to design the two-level PDE invariance principle. The proposed iMOOE leverages a fusion network to approximate $F(\cdot)$ and a set of neural operator experts to approximate $\{\sigma_i(\mathbf{u}, \mathbf{x})\}_{i=1}^K$.

Secondly, as observed in classical PDE discovery literature such as SINDy (Brunton et al., 2016) and PDE-FIND (Rudy et al., 2017), PDE systems of interests can be naturally represented in a few spatial operators. These PDE discovery methods resort to SVD-based linear combination of a few terms in the fixed operator library. This indeed motivates us to design the operator invariance principle, and also justifies the compositional structure of operator learning.

Thirdly, we justify the composition of neural operators (i.e. iMOOE's architecture design) can approximate arbitrary continuous operators according to the universal approximation theorem proved in (Kovachki et al., 2023). As illustrated in Section 9.3 and Figure 16 in (Kovachki et al., 2023), neural operators that can approximate continuous operators should possess three modules: i) Mapping input to the first hidden representation by a local operator. ii) Mapping each hidden representation to next by a non-local integral kernel operator. iii) Mapping the last hidden representation to output by a local operator. In practical implementation, i) and iii) can be realized by a local neural network like MLP, while ii) should be realized by a global neural network like transformer or FNO layers. The proposed composition of neural operators realizes i) and ii) by concatenating the last hidden representations of multiple neural operators, and realizes iii) by the fusion network. Hence, iMOOE architecture design can also satisfy the universal approximation theorem of operators.

# C  RELATED WORK

## C.1  SPATIOTEMPORAL PDE DYNAMICS FORECASTING

Deep learning-based dynamics forecasting centers around developing diverse neural operators to decipher the unknown time-dependent PDE systems. This line of research has been extensively leveraged to reason a wide scope of real-world spatiotemporal dynamics, like atmospheric circulation (Pathak et al., 2022), ocean wave (Cui et al., 2025), turbulent fluid (Xing et al., 2024), and power system transient (Cui et al., 2023). Their innovations come from either new operator architectures or more robust autogressive training methods, which are aimed at addressing a range of open issues in PDE forecasting, including solving parametric PDEs (Takamoto et al., 2023; Cho et al., 2024), full-field reconstruction from sparse observations (Shysheya et al., 2024; Li et al., 2024b), irregular geometries (Li et al., 2023c; Wu et al., 2024a) or long temporal process stability (Lippe et al., 2023; Rühling Cachay et al., 2023). However, most of these works do not highlight the zero-shot OOD generalizable forecasting, which is a significantly crucial problem for two reasons: i) The unseen

OOD scenarios can always occur in real-world PDE dynamics prediction, owing to the ubiquitous distribution shifts of forecasting contexts, encompassing system parameters, external forcing functions, initial conditions and sampling conditions. ii) As procuring abundant dynamics trajectories to learn domain-transferable representations is expensive, many PDE dynamics forecasting methods are cultivated in low-data regime. In this sense, how to generalize across diverse OOD environments with limited training data is of great importance. Although several studies have explored the potential of meta-learning (Kirchmeyer et al., 2022; Kassaï Koupaï et al., 2024) or parameter conditioning (Takamoto et al., 2023; Gupta & Brandstetter, 2022) methods in OOD forecasting, their zero-shot generalization capability remains lacking since they can not expose the truly fundamental invariance in PDE dynamical systems. Another drawback is that they need to carry out ad-hoc modifications to current neural operator architectures. To remedy them, we elucidate the physical invariance principle of PDE dynamics from two perspectives, and then develop a Mixture-of-Expert (MoE)-based architecture which can delicately integrate existing operator learning methods to capture PDE invariance in a plug-and-play manner.

Note that the MoE-based architecture (Dai et al., 2024) has been extensively employed in Large Language Models (LLM) to increase the representation capacity and knowledge density without sacrificing the inference speed. Few neural PDE solvers based on spatial domain decomposition (Hao et al., 2023; Chalapathi et al., 2024) also borrow this parallel structure to improve the computational efficiency for large-scale PDEs. It shares the same spirit with the finite element method, as each expert is assigned to calculate on a sub-domain and coordinating these experts can behave well on complex geometries. In contrast, our proposed mixture of operator expert architecture is aimed at capturing the domain-invariant operators for zero-shot generalizable forecasting. Another difference from LLM on MoE usage is that experts in LLM are sparsely activated according to the routed token, whereas MoE in neural PDE is always dense as each expert should account for sub-operators or sub-domains.

## C.2 Invariant Learning for OOD Generalization

Invariant learning (Arjovsky et al., 2019; Liu et al., 2021) is an effective paradigm to boost OOD generalization performance. It aims to discover invariant representations that can possess sufficient information to predict targets and elicit equal risks across various (unseen) environments. There exist two open issues in invariant learning: i) how to prescribe the domain-specific invariance principle for different learning problems; ii) how to design the effective OOD objectives to estimate the defined invariance on limited training contexts. Existing works strive to address these challenges from different aspects, such as feature learning (Chen et al., 2023a), multi-objective optimization (Chen et al., 2023b), architecture alignment (Li et al., 2023a), information bottleneck (Ahuja et al., 2021) and gradient consistency (Rame et al., 2022). These research outcomes have been successfully applied to vision recognition (Li et al., 2023a), molecule prediction (Chen et al., 2022; Li et al., 2022), pedestrian motion forecasting (Liu et al., 2022) and time series analysis (Liu et al., 2024). However, the efficacy of invariant learning for PDE dynamics forecasting is still under-explored. To bridge this gap in this work, we propose to unleash its power by fostering an iMOOE architecture and optimizing it by a frequency-enriched risk equality loss, both of which can help to capture the complete PDE dynamics invariance.

## C.3 Physical Invariance Learning

Incorporating physical prior knowledge into deep learning is a valid way to improve the generalization capacity, data efficiency as well as the physical consistency of produced predictions (Yu & Wang, 2024). In light of this, a line of related research focus on imposing domain-specific physics knowledge which remains invariant in PDE dynamical systems, for the sake of better accuracy and OOD robustness of dynamics reasoning. These physical invariance can involve symmetries (Wang et al., 2021), conservation laws (Huang et al., 2024), exact physics models (Holt et al., 2024) or basis function dictionaries (Mouli et al., 2024). In this work, we propose a two-level invariance principle for PDE dynamics inspired by the formation process of PDE laws and useful operator splitting method. Such PDE invariance can be deemed as a kind of prior physical knowledge, which need to be digged out by physics-informed invariant learning.

## C.4 Practical value of Zero-shot OOD Dynamics Forecasting

In the scope of both parametric PDE simulation and real-world PDE-governed physical dynamics forecasting, the zero-shot OOD generalization is a ubiquitous and urgent issue. i) In many industrial manufacturing fields which require high-intensity PDE calculation, such as electromagnetic simulation (Huang et al., 2022) and airfoil design (Wu et al., 2024b), PDE parameters are ever-changing due to the varying material properties and ambient factors. It is also hard to acquire valuable test-time trajectories for each new physical environment. Thus the zero-shot OOD simulation is highly demanded. ii) In many spatiotemporal physical dynamics forecasting fields such as weather and climate prediction (Bodnar et al., 2025), there always exist unforeseen dynamics patterns in meterological variables due to the chaotic nature of systems and unpredictable human activities. It is impossible to collect abundant training contexts which can cover all the unforeseen test scenarios. It is also computational expensive to fine-tune the weather foundation model (Bodnar et al., 2025) for the hourly or daily inference. Thus the zero-shot OOD forecasting is greatly significant.

## D Multi-Environment Dataset Details

Unless otherwise stated, the experiments conducted in this work follow the same data setting during the training and test stage: i) Training data: 16 environments with 64 trajectories per environment. ii) Test data: 16 environments with 8 trajectories per environment. All OOD forecasting experiments are executed in zero-shot settings, without any test-time samples for model fine-tuning or adaptation. Below, we clarify the multi-context state trajectory generation method on five two-dimensional PDE dynamical systems. We assume the boundary conditions (BCs) are fixed (e.g. periodic BC) for each PDE system, so that BCs are not regarded as environment variables.

**Diffusion-Reaction (Takamoto et al., 2022).** The underlying DR equation is presented as:

$$\partial_t u = D_u \partial_{xx} u + D_u \partial_{yy} u + \left(u - u^3 - k - v\right), \tag{10}$$
$$\partial_t v = D_v \partial_{xx} v + D_v \partial_{yy} v + (u - v). \tag{11}$$

$u, v$ denote the concentrations of activator and inhibitor respectively. The spatiotemporal domain is $(\mathbf{x}, t) \in [0, 2]^2 \times [0, 20]$. At the initial state, two objects are randomly localized into six $0.2 \times 0.2$ squares. Two diffusion coefficients $D_u, D_v$ and one reaction coefficient $k$ are assigned to construct different physical contexts. The physical parameters of training trajectories are drawn from $D_u \in [1e\text{-}3, 2e\text{-}3]$, $D_v \in [5e\text{-}3, 1e\text{-}2]$, $k \in [5e\text{-}3, 1e\text{-}2]$, while the OOD test parameters are fetched from $D_u \in [2e\text{-}3, 3e\text{-}3]$, $D_v \in [1e\text{-}2, 1.5e\text{-}2]$, $k \in [1e\text{-}2, 1.5e\text{-}2]$. We utilize past 10 steps sequence to forecast future 11 steps states, i.e. $H = W = 10$ and $N_t = 21$.

**Navier-Stokes (Li et al., 2021).** The vorticity-type incompressible NS equation is presented as:

$$\partial_t \omega = -\mathbf{u} \cdot \nabla \omega + \nu \Delta \omega + 0.1 \left(\sin(w\pi(x + y)) + \cos(w\pi(x + y))\right), \nabla \cdot \mathbf{u} = 0. \tag{12}$$

$\omega, \mathbf{u}$ denote the velocity field and fluid vorticity. The spatiotemporal domain is $(\mathbf{x}, t) \in [0, 1]^2 \times [0, 50]$. The initial vorticity are produced from a normal Gaussian random field. The viscosity coefficient $\nu$ and the frequency coefficient $w$ in the forcing term can be employed to generate diverse physical environments. We simulate training sequences using $\nu \in [1e\text{-}5, 1e\text{-}3]$ and OOD test sequences using $\nu \in [5e\text{-}6, 8e\text{-}6] \cup [1.2e\text{-}3, 2e\text{-}3]$ with a fixed $w = 2$. We utilize previous 10 steps trajectories to forecast future 21 steps vorticity, i.e. $H = W = 10$ and $N_t = 31$.

**Burgers (Hao et al., 2024b).** The coupled BG equation is presented as:

$$\partial_t \mathbf{u} = -\mathbf{u} \cdot \nabla \mathbf{u} + \nu \Delta \mathbf{u}. \tag{13}$$

$\mathbf{u}$ denotes the fluid velocity field. The spatiotemporal domain is $(\mathbf{x}, t) \in [0, 64]^2 \times [0, 1]$. Identical to (Hao et al., 2024b), we also adopt the same sine and cosine functions over the spatial domain to generate initial conditions. The viscosity coefficient $\nu$ is employed to produce diverse forecasting scenarios. The training sequences are simulated from $\nu \in [5e\text{-}3, 5e\text{-}2]$, whereas the OOD test sequences are simulated from $\nu \in [2.5e\text{-}3, 4e\text{-}3] \cup [6e\text{-}2, 1e\text{-}1]$. The past 10 steps series are utilized to forecast future 11 steps velocity, i.e. $H = W = 10$ and $N_t = 21$.

**Shallow-Water (Takamoto et al., 2022).** The hyperbolic SW equation is presented as:

$$\partial_t h + \partial_x hu + \partial_y hv = 0, \tag{14}$$

$$\partial_t hu + \partial_x \left( u^2 h + \frac{1}{2} g_r h^2 \right) = -g_r h \partial_x b, \tag{15}$$

$$\partial_t hv + \partial_y \left( v^2 h + \frac{1}{2} g_r h^2 \right) = -g_r h \partial_y b. \tag{16}$$

$u$, $v$ denote the velocities along the horizontal and vertical axis. $h$ denotes the water depth and $b$ is a spatially varying bathymetry. $hu$, $hv$ can be perceived as the directional momentum components. $g_r$ indicates the acceleration of gravity. The spatiotemporal domain is $(\mathbf{x}, t) \in [0, 5]^2 \times [0, 1]$. Akin to (Takamoto et al., 2022), the initial conditions are shaped as 2D radial dam breaks. We take their initial radius as physical parameters to construct data contexts. The training series are fetched from radius within $[0.3, 0.63]$, and the OOD test series are obtained from radius within $[0.63, 0.7]$. The prior 10 steps series are utilized to forecast the water depth of future 11 steps, i.e. $H = W = 10$ and $N_t = 21$.

**Heat-Conduction (Hao et al., 2024b).** The HC equation with a varying heat source is presented as:

$$\partial_t u = \nabla(a(\mathbf{x})\nabla u) + A \sin(m_1 \pi x) \sin(m_2 \pi y) \sin(m_3 \pi t). \tag{17}$$

$u$ denotes the temperature field over the spatiotemporal domain $(\mathbf{x}, t) \in [0, 1]^2 \times [0, 5]$. Similar to (Hao et al., 2024b), the coefficient function $a(x)$ is stipulated as a exponential Gaussian random field. The external forcing terms are altered to generated various physical contexts. We specifically vary three frequency coefficients $m_1$, $m_2$, $m_3$ of heat sources and keep the amplitude $A = 200$. The training temperature fields are produced by $m_1, m_3 \in [1, 2]$, $m_2 \in [5, 10]$, and the OOD test fields stem from $m_1, m_3 \in [2, 3]$, $m_2 \in [10, 15]$. We utilize past 10 steps fields to forecast the temperature of future 11 steps, i.e. $H = W = 10$ and $N_t = 21$.

# E  MORE ANALYSIS ON MIXTURE OF OPERATOR EXPERT ARCHITECTURE

## E.1  DISPARATE MOE USAGE IN IMOOE AND LFMS

The key difference lies in during the forward pass, LFMs (Dai et al., 2024; Liu et al., 2025; Shi et al., 2025) need to selectively activate a sparse number of FFN experts, while iMOOE stands for a dense version of MoE which should aggregate the output of all neural operator experts by the designed fusion network. Commonly, LFMs require a huge number of experts to express the fine-grained and specialized knowledge in pretraining data corpus, and their performance on downstream tasks can benefit from the large capacity of specialized experts. However, as for PDE invariance learning, the invariant knowledge is prescribed as the composition of a few number of invariant operators, since real-world PDE dynamical systems often consist of a small set of physical processes (Rudy et al., 2017). Accordingly, iMOOE can capture the underlying PDE law by only a few number of operator experts. Besides, LFMs usually enable each FFN expert to represent distinct knowledge by the load balance loss (Dai et al., 2024). While iMOOE leverages the proposed mask diversity loss to adaptively select different sets of spatial derivatives for expert input, which can explicitly enforce individual operator experts to express distinct physical processes.

## E.2  DETAILED EXPLANATIONS ON THE EFFECT OF THE NUMBER OF OPERATOR EXPERTS

The mixture of operator expert architecture is specifically designed to closely align with the proposed two-level PDE invariance principle. In Table 6, we can observe it is not strict that iMOOE's zero-shot OOD capability can constantly promote with the increase of the expert number $K$ for two reasons: i) *Overfitting risk*. A large $K$ will increase iMOOE's model complexity. When $K$ is overly large but the operator invariance is not that complex, such as $K = 4$ for DR dynamics, iMOOE is likely to overfit to the limited training domains (i.e. 16 training environments with to 1024 DR trajectories). It can diminish the accuracy and robustness of captured PDE invariance. ii) *Representation redundancy*. Real-world PDE systems are usually composed by a few number of physical processes, such as the DR system only contains a Laplacian operator and a reaction function. A overly large $K$ could render the representations of these FNO experts redundant to each other. For example,

when we input second-order derivatives $[u_{xx}, v_{xx}, u_{yy}, v_{yy}, u_{xy}, v_{xy}]$ to four FNO experts to learn DR dynamics, their actual learned masks are $\mathbf{m}_1 = [0, 1, 1, 0, 1, 1]$, $\mathbf{m}_2 = [0, 1, 1, 1, 0, 1]$, $\mathbf{m}_3 = [1, 0, 0, 0, 1, 1]$, $\mathbf{m}_4 = [0, 0, 1, 0, 0, 0]$. We can observe that the first and second expert behaves very similarly to each other, and the fourth expert is unnecessary since its behavior can be covered by other three experts. Thus $K = 3$ can perform better than $K = 4$ as shown in Table 6.

# F ADDITIONAL RESULTS

## F.1 ABLATION STUDY

### F.1.1 EFFECT OF FREQUENCY-ENRICHED LOSS IN EQ. 8

We investigate the benefits of the proposed frequency enrichment loss $\mathcal{L}_{freq}$ for PDE invariance learning. We utilize the simulated DR data and real-word SST, SSE data to validate the improved forecasting generalizability induced by additional regularization on high-frequency representations. Diffusion and reaction coefficients can dictate the distribution of frequency patterns in DR evolutions. SSE contains high-frequency short waves which could be caused by the nonlinear surface features like wave breaking fronts, sharp crests and bound harmonics. SST contains high-frequency modes due to the ocean advection and vertical processes such as upwelling. Apart from nMESE and fRMSE metrics, we also present the forecasting errors within different frequency bands in Table 7, 8. "Low", "Mid", "High" denote non-overlapped ranges of wavenumber $\xi$: $\xi_{\text{low}} \in [0, 4]$, $\xi_{\text{mid}} \in [5, 12]$, $\xi_{\text{high}} \in [13, \xi_{max}]$. When equipped with high-frequency augmentation, the ID/OOD nMSE can drop by $24.38\%$ and $25.00\%$, and ID/OOD fRMSE can decrease by $10.20\%$ and $12.32\%$ on DR data. It also improves nMSE by $9.22\%$ and $8.98\%$, fRMSE by $7.03\%$ and $7.33\%$ for SST, SSE data. Notably, improving high-frequency feature learning can also enhance the OOD accuracy on both low-frequency and mid-frequency patterns. Such ID/OOD promotion verifies the necessity of the proposed frequency-enriched objective, which can mitigate the spectral bias of neural operators and help to capture the complete PDE invariance from the spectral domain.

Table 7: Ablation results of frequency enrichment loss on DR data.

| Methods | ID | | | | | OOD | | | | |
|---|---|---|---|---|---|---|---|---|---|---|
| | nMSE | fRMSE | | | | nMSE | fRMSE | | | |
| | | Low | Mid | High | Total | | Low | Mid | High | Total |
| w/o $\mathcal{L}_{freq}$ | 6.81e-3 | 3.39e-3 | 1.17e-3 | 3.31e-4 | 1.02e-3 | 5.64e-2 | 9.29e-3 | 1.25e-3 | 4.42e-4 | 2.03e-3 |
| w/ $\mathcal{L}_{freq}$ | **5.15e-3** | **2.96e-3** | **1.09e-3** | **3.03e-4** | **9.16e-4** | **4.23e-2** | **7.92e-3** | **1.17e-3** | **4.20e-4** | **1.78e-3** |

Table 8: Ablation results of frequency enrichment loss on real-word ocean dynamics data.

| Methods | SST | | | | | SSE | | | | |
|---|---|---|---|---|---|---|---|---|---|---|
| | nMSE | fRMSE | | | | nMSE | fRMSE | | | |
| | | Low | Mid | High | Total | | Low | Mid | High | Total |
| w/o $\mathcal{L}_{freq}$ | 5.64e-1 | 1.29e-1 | 3.55e-2 | 9.24e-3 | 3.70e-2 | 1.67e-2 | 3.03e-3 | 2.98e-3 | 3.01e-3 | 3.00e-3 |
| w/ $\mathcal{L}_{freq}$ | **5.12e-1** | **1.24e-1** | **3.06e-2** | **8.47e-3** | **3.44e-2** | **1.52e-2** | **2.10e-3** | **2.74e-3** | **2.94e-3** | **2.78e-3** |

### F.1.2 EFFECT OF PRE-CALCULATED DERIVATIVE SELECTION

We investigate the effect of input spatial derivative selection designed in Section 3.1. Such design incorporates certain orders of pre-calculated spatial derivatives into each operator expert input and a specific mask diversity loss which can encourage experts to represent distinct operators. We report the influence of this derivative selection design in Table 9. We can find that especially for the real-world SST changing dynamics which are complex and hard to capture, the prior derivative input can make it easier and more accurate to discover SST's physical law. (Li et al., 2024a) consistently validated that introducing additional spatial derivatives can improve neural PDE learning. We conduct a further analysis on this design as follows:

i) *Effect of mask diversity loss $\mathcal{L}_{mask}$.* We set $\lambda_{mask} = 0$ and feedforward all pre-computed derivatives to each expert. The OOD nMSE and fRMSE results on DR data are $4.78e - 2$ and $1.89e - 3$,

leading to 13.0% and 6.18% degradation versus standard iMOOE. We find masks learned by two experts are similar to each other, which hinders them from representing distinct invariant operators.

ii) *Effect of derivative types.* As BG equation contains first-order and second-order derivatives, we take both of them as prior input for vanilla iMOOE, and each mask learns to adaptively select the needed derivatives for its coupled operator expert. But when we just input first-order derivatives, OOD nMSE and fRMSE increase to $1.16e - 2$ and $3.85e - 3$, with 7.41% and 0.52% degradation. This verifies that prior second-order derivatives can improve learning efficiency for BG systems.

iii) *Actual learned mask vectors* $\mathbf{m}$. When learning on DR data, we feed second-order derivatives $[u_{xx}, v_{xx}, u_{yy}, v_{yy}, u_{xy}, v_{xy}]$ to two experts in iMOOE, and their actual learned mask is $\mathbf{m}_1 = [0, 0, 1, 1, 1, 0]$, $\mathbf{m}_2 = [1, 1, 1, 0, 0, 1]$. As there are many operator splitting methods for DR equation (e.g. dividing into diffusion and reaction terms is just one of them), iMOOE can learn a suitable splitting way via learning operator invariance from limited data.

Table 9: Ablation results of input spatial derivative selection.

| Methods | DR | | BG | | SST | |
|---|---|---|---|---|---|---|
| | nMSE | fRMSE | nMSE | fRMSE | nMSE | fRMSE |
| w/o derivative selection | 4.95e-2 | 1.95e-3 | 1.13e-2 | 3.94e-3 | 6.07e-1 | 3.82e-2 |
| w/ derivative selection | 4.23e-2 | 1.78e-3 | 1.08e-2 | 3.83e-3 | 5.12e-1 | 3.44e-2 |
| Degradation ↓ | 17.02% | 9.55% | 4.63% | 2.87% | 18.55% | 11.05% |

### F.1.3 Effect of the Choice of Fusion Network

We verify that properly choosing the type of expert fusion methods (presented in Section 3.1) is crucial to learn the accurate PDE invariance. We can determine the type of fusion network in light of prior physical knowledge on PDE systems. To focus on this network structure study, we abandon additional multi-environment invariance training. We take DR and NS systems for comparison and provide OOD results in Table 10. We can find that for linear PDE systems such as DR, which holds a simple additive relationship between the diffusion operator and reaction function, simply summing up the outputs of operator experts is a better fit. But for strongly non-linear PDE systems like NS, which include complex operator multiplication, we should impose an extra fusion network and let it learn how to integrate expert outputs to capture the non-linear PDE law.

Table 10: Ablation results of the choice of two types of fusion methods.

| Expert Composition Methods | DR | | NS | |
|---|---|---|---|---|
| | nMSE | fRMSE | nMSE | fRMSE |
| Linear fusion by simple addition | **5.80e-2** | **2.02e-3** | 4.82e-1 | 6.32e-2 |
| Non-linear fusion by extra network | 6.46e-2 | 3.28e-3 | **3.76e-1** | **5.54e-2** |

### F.1.4 Effect of Two Environment Partition Methods

As mentioned in Section 3.2, dividing the training environments based on autoregressive time steps can further boost the outcomes of PDE invariance learning. We adopt two fluid dynamics datasets to verify the benefit of this step-wise partition method in addition to common parameter-based division. During the fluid evolution, state variations on two consecutive time steps are quite distinct, but the physics transition law between these two steps remain invariant. Therefore, we can regard each autoregressive step as a unique context. In Table 11, we present the effect of two environment partition methods. We can see that combining two partition methods together can realize the best ID/OOD performance, since it can enhance the diversity of training environments and improve the robustness of learned PDE invariance representations. Besides, parameter-based partition performs moderately better than step-wise partition, as different physical parameters can lead to more distinct PDE trajectories, such as the Reynold number in NS is a decisive factor to distinguish laminar or turbulent flow.

Table 11: Effect of two environment partition methods on fluid forecasting.

| Partition methods | NS | | | | BG | | | |
| | ID | | OOD | | ID | | OOD | |
| | nMSE | fRMSE | nMSE | fRMSE | nMSE | fRMSE | nMSE | fRMSE |
|---|---|---|---|---|---|---|---|---|
| Only parameters | 7.11e-2 | 1.50e-2 | 3.41e-1 | 5.49e-2 | 1.48e-3 | 1.17e-3 | 1.11e-2 | 3.94e-3 |
| Only time steps | 7.39e-2 | 1.52e-2 | 3.58e-1 | 5.52e-2 | 1.57e-3 | 1.19e-3 | 1.18e-2 | 4.01e-3 |
| Parameters+time steps | **6.49e-2** | **1.38e-2** | **3.12e-1** | **5.36e-2** | **1.20e-3** | **1.10e-3** | **1.08e-2** | **3.83e-3** |

### F.1.5 EFFECT OF LINEAR LOSS SCHEDULING

According to previous invariant learning implementation (Krueger et al., 2021), the linear scheduling scheme is an effective and canonical way to impose the risk equality loss $\mathcal{L}_{inv}$ on neural networks. To probe its effect on PDE invariance learning, we compare the performance of the MOOE model with fixed $\mathcal{L}_{inv}$ or linearly added $\mathcal{L}_{inv}$ in Table 12. Concretely, "fixed" means $\mathcal{L}_{inv}$ keeps at 0.001 during the whole training procedure. "Linearly scheduled" indicates $\mathcal{L}_{inv}$ is zero during the initial 175 epochs, then linearly increases to 0.001 during the intermediate 150 epochs, and finally stays at 0.001 during the last 175 epochs. The main distinction between these two schemes lies in whether executing traditional empirical risk minimization (ERM) training by the maximal prediction loss $\mathcal{L}_{pred}$ during the initial pretraining stage of 175 epochs. Prior invariant learning works (Chen et al., 2023a; Zhang et al., 2022) claim that native ERM pretraining can help to gain rich data representations at the beginning. Invariant learning can be deemed as a certain way to filter out the domain-generalizable representations. We verify its effect on DR dynamics as shown in Table 12. We can find that compared to fixing $\mathcal{L}_{inv}$ from scratch, linearly imposing $\mathcal{L}_{inv}$ on MOOE can lead to better ID/OOD forecasting accuracy and lower error variance across test environments. It reflects that linear scheduling scheme can be better way to conduct the PDE invariance learning objective when training on diverse physical environments.

Table 12: Ablation results of linear invariant loss scheduling on DR data.

| Methods | ID | | | | OOD | | | |
| | nMSE | | fRMSE | | nMSE | | fRMSE | |
| | Mean | Std | Mean | Std | Mean | Std | Mean | Std |
|---|---|---|---|---|---|---|---|---|
| MOOE+fixed $\mathcal{L}_{inv}$ | 5.61e-3 | 4.93e-3 | 1.04e-3 | 3.91e-4 | 5.80e-2 | 7.59e-2 | 2.02e-3 | 4.76e-4 |
| MOOE+linearly scheduled $\mathcal{L}_{inv}$ | **5.06e-3** | **4.84e-3** | **9.23e-4** | **3.86e-4** | **4.93e-2** | **5.77e-2** | **1.87e-3** | **4.06e-4** |

### F.2 APPLICABILITY TO DIVERSE DYNAMICAL SYSTEM FORECASTING

In this section, we demonstrate the proposed physics-informed invariant learning method iMOOE can be easily extended to a wide variety of dynamics forecasting scenarios, apart from the 2D PDE systems on regular grids. In the following, we validate iMOOE's zero-shot OOD forecasting performance on neural ODE systems, 3D fluid dynamics and real-world time series. We can adapt iMOOE to these diverse dynamics by replacing the expert backbone with task-specific architectures.

### F.2.1 APPLICATION TO ODE-GOVERNED DYNAMICS FORECASTING

We verify iMOOE's zero-shot OOD capability on neural ODE systems, as prior meta-learning-based methods such as CoDA (Kirchmeyer et al., 2022) and GEPS (Kassaï Koupaï et al., 2024) are also extended to ODE-governed dynamics forecasting. We conduct simulation on a typical ODE system called damped and driven pendulum equation, as shown in Appendix B.1 of (Kassaï Koupaï et al., 2024). We utilize past 10-step states to forecast the pendulum motion angle of future 41 steps. The time horizon of collected trajectories is $[0, 25]$ and $N_t = 51$. We construct 16 ID training domains and 8 OOD test domains by randomly drawing four ODE parameters from the ID/OOD ranges given in Table 13. Identical to CoDA and GEPS, a 4-layer MLP network with 64 hidden dimension is taken as the backbone for operator experts of iMOOE. The pre-calculated spatial derivatives and mask diversity loss are discarded since they are unnecessary for ODE simulation. We report OOD

forecasting performance of zero-shot iMOOE and few-shot CoDA, GEPS in Table 14. iMOOE can achieve 10.4% and 12.66% decrease on nMSE and fRMSE versus GEPS. This may stem from their difference on discovering physical invariance. Specifically, iMOOE explicitly prescribes the two-level invariance principle and directly captures it via the proposed physics-informed invariant learning. While hypernetwork-based meta-learning methods like CoDA, GEPS estimate such invariance by implicitly operating in the network parameter space without any physical guidance.

Table 13: ID/OOD parameter ranges of pendulum system for environment generation.

| Parameters | ID Range | OOD Range |
|---|---|---|
| Damping coefficient $\alpha$ | [0.1,0.2] | [0.2,0.3] |
| Natural frequency $\omega_0$ | [0.5,1.0] | [1.0,1.5] |
| Forcing frequency $\omega_f$ | [0.3,0.6] | [0.6,0.9] |
| Forcing amplitude $F$ | [0.1,0.2] | [0.2,0.3] |

Table 14: OOD forecasting results on ODE-governed pendulum dynamics.

| Models | nMSE | fRMSE |
|---|---|---|
| CoDA | 5.31e+0 | 5.96e-2 |
| GEPS | 2.50e+0 | 5.45e-2 |
| iMOOE | **2.24e+0** | **4.76e-2** |

### F.2.2 APPLICATION TO 3D FLUID DYNAMICS FORECASTING

Apart from the typical 2D PDE dynamics, we also demonstrate iMOOE's zero-shot OOD performance on 3D PDE systems. We employ the 3D compressible Navier-Stokes equation in PDEBench (Takamoto et al., 2022) and construct ID and OOD scenarios using the same method in Appendix D. Specifically, for shear and bulk viscosity coefficients, we still randomly draw their values from the ID parameter range $[1e-5, 1e-3]$ and OOD range $[5e-6, 8e-6] \cup [1.2e-3, 2e-3]$. The Mach number is kept as 1.0. The number of ID training and OOD test domains as well as their data volume are also identical to setups in Appendix D. The size of 3D spatial domain is $32 \times 32 \times 32$, and the past 10-step velocity field sequences are provided to forecast the future 11-step states. FNO3d is utilized as the backbone of operator experts for iMOOE3d. We present the 3D OOD dynamics forecasting results in Table 15. We observe that iMOOE can attain 17.36% and 39.26% decrease on nMSE and fRMSE compared to previous 3D neural operators. Such results further validate the effectiveness of proposed physics-guided PDE invariance learning to more complex 3D PDE dynamics.

### F.2.3 APPLICATION TO REAL-WORLD TIME SERIES PREDICTION

We further validate iMOOE's OOD capability on real-world time series prediction. Such time series dynamics are hard to be directly parsed by ODE or PDE laws. We leverage the Electricity Transformer Temperature (ETT) data (Zhou et al., 2021) and follow the data split setting in (Zhou et al., 2021). The changing dynamics of Oil Temperature (OT) is hard to decipher. OT dynamics are associated with exogenous covariates like electricity load. The task is to predict future 96-step OT values given lookback 512-step OT and six auxiliary power load sequences. To implement iMOOE on this task, we borrow the Moirai-MoE (Liu et al., 2025) as backbone to approximate the invariant knowledge in ETT dynamics. The proposed frequency-augmented invariant learning objective is utilized to fine-tune Moirai-MoE. As temporal distribution shifts are ubiquitous in time series domain (Kim et al., 2021), we simply deem each segmented ETT trajectory as an independent environment. Consequently, iMOOE can attain 25.29% and 13.59% growth on nMSE and fRMSE compared to native Moirai-MoE as reported in Table 16.

Table 15: ID/OOD forecasting results on 3D NS dynamics.

| Models | nMSE | | fRMSE | |
|---|---|---|---|---|
| | ID | OOD | ID | OOD |
| UNet3d | 1.67e+0 | 1.92e+0 | 3.09e-1 | 5.16e-1 |
| FNO3d | 3.83e-1 | 1.88e+0 | 4.99e-2 | 1.84e-1 |
| VCNeF3d | 1.69e-1 | 6.97e-1 | 5.88e-2 | 1.35e-1 |
| iMOOE3d | **1.19e-1** | **5.76e-1** | **2.11e-2** | **8.20e-2** |

Table 16: OOD forecasting results on ETT time series dynamics.

| Models | nMSE | fRMSE |
|---|---|---|
| Informer | 3.17e-1 | 1.07e-2 |
| Moirai-MoE | 5.26e-2 | 4.93e-3 |
| iMOOE | **3.93e-2** | **4.26e-3** |

### F.2.4 APPLICATION TO PDE DYNAMICS ON IRREGULAR SPATIAL DOMAIN

We leverage the Airfoil forecasting benchmark on non-uniform grid proposed in OFormer (Li et al., 2023b), where the underlying grid is divided by highly irregular triangular meshes. This Airfoil dataset contains 1000 training and 100 testing sequences with different inflow speed (Mach number) and angles of attack. The objective is to jointly predict future 22 steps velocity, density and pressure using past 4 steps states. One of the key advantages of iMOOE is the elegant compatibility across various neural operators. Hence, iMOOE can integrate neural operators that are able to handle irregular spatial domains or unstructured meshes as well, such as OFormer (Li et al., 2023b), Geo-FNO (Li et al., 2023c) and VCNeF (Hagnberger et al., 2024). We can easily enable iMOOE on irregular geometries by two tiny adaptations: i) The prior spatial derivatives on irregular grid are calculated by the finite element method. ii) The frequency enrichment loss relying on fast fourier transform on uniform grid is discarded. In Table 17, we present that when equipped with the proposed iMOOE method, three neural operators can exhibit better OOD forecasting accuracy on flows around Airfoil. We randomly showcase a OOD flow velocity forecast in Fig. 15, 16.

Table 17: OOD forecasting results on irregular spatial domains.

| Operators | Variants | nMSE | RMSE |
|---|---|---|---|
| VCNeF | Naive | 5.45e-2 | 2.22e-1 |
| | **+iMOOE** | **5.08e-2** | **1.98e-1** |
| Geo-FNO | Naive | 5.15e-2 | 2.16e-1 |
| | **+iMOOE** | **4.35e-2** | **1.88e-1** |
| OFormer | Naive | 4.95e-2 | 2.04e-1 |
| | **+iMOOE** | **4.14e-2** | **1.79e-1** |

### F.3 MORE RESULTS ON SENSITIVITY ANALYSIS

By hyperparameter tuning, we empirically find the fixed setting $\lambda_{pred} = 1.0, \lambda_{inv} = 0.001, \lambda_{freq} = 0.1, \lambda_{mask} = 0.001$ can perform well on both simulated and real-world physical dynamics data. As the effect of $\mathcal{L}_{mask}$ has been discussed in Appendix F.1.2, we investigate iMOOE's sensitivity to different $\lambda_{inv}$ and $\lambda_{freq}$ values. OOD results in Table 18 reflect that we should assign moderate values to $\lambda_{inv}$ and $\lambda_{freq}$ for satisfactory outcomes.

i) Sensitivity to invariance loss weight $\lambda_{inv}$. For a smaller $\lambda_{inv} = 0.0001$, it diminishes the power of $\mathcal{L}_{inv}$ to capture the physical invariance and renders iMOOE overfit to training environments. For a larger $\lambda_{inv} = 0.01$, the degradation stems from the intrinsic conflict between $\mathcal{L}_{inv}$ and $\mathcal{L}_{pred}$ according to Section 3 in REx (Krueger et al., 2021). REx claims that overly minimizing the variance of errors across training domains can increase the error of the best-performing domain.

ii) Sensitivity to frequency loss weight $\lambda_{freq}$. For a smaller $\lambda_{freq} = 0.01$, the generalization errors caused by high-frequency pitfalls can not be mitigated. For a larger $\lambda_{freq} = 1.0$, the high-frequency modes are over-optimized but the dominant low-frequency modes are not learned well.

Table 18: Influence of loss weights in Eq. 9.

| Loss | $\lambda_{inv}$ | $\lambda_{freq}$ | nMSE | | fRMSE | |
|---|---|---|---|---|---|---|
| | | | ID | OOD | ID | OOD |
| $\mathcal{L}_{inv}$ | 0.01 | 0.1 | 5.58e-3 | 5.15e-2 | 9.26e-4 | 1.94e-3 |
| | 0.0001 | 0.1 | 5.40e-3 | 5.24e-2 | 9.12e-4 | 1.98e-3 |
| $\mathcal{L}_{freq}$ | 0.001 | 1.0 | 5.75e-3 | 5.04e-2 | 9.30e-4 | 1.95e-3 |
| | 0.001 | 0.01 | 5.42e-3 | 4.89e-2 | 9.18e-4 | 1.88e-3 |
| Ours | 0.001 | 0.1 | **5.10e-3** | **4.23e-2** | **9.16e-4** | **1.78e-3** |

### F.4 FURTHER COMPARISON WITH META-LEARNING-BASED METHODS

We first clarify how to adapt meta-learning-based baselines including CoDA (Kirchmeyer et al., 2022) and GEPS (Kassaï Koupaï et al., 2024) to zero-shot OOD forecasting. Both CoDA and GEPS separate their network parameter space into domain-invariant and domain-specific parts. Domain-specific parameters need to be independently trained within each unique environment and require few-shot adaptation. When applied to zero-shot OOD testing, domain-invariant parameters can keep freezing, while domain-specific parameters including $\theta^e$ in CoDA and $\mathbf{c}^e$ in GEPS are initialized by averaged parameters over diverse training domains (e.g. $\bar{\mathbf{c}}_{tr} = \frac{1}{|\mathcal{E}_{tr}|} \sum_{e=1}^{|\mathcal{E}_{tr}|} \mathbf{c}^e$). This test-time initialization method for domain-specific parameters is directly borrowed from GEPS (Kassaï Koupaï et al., 2024), as stated in its last paragraph of Section 4.1.

Furthermore, we implement CoDA and GEPS through their vanilla few-shot adaptation manner and our zero-shot inference setting. For few-shot setting, as claimed in Section 5.2 of both CoDA (Kirchmeyer et al., 2022) and GEPS (Kassaï Koupaï et al., 2024), we draw only one PDE trajectory from each unseen test environment to finetune the domain-specific network parameters. In Table 19, we present OOD forecasting outcomes of zero-shot iMOOE and zero/few-shot CoDA and GEPS. It is apparent that iMOOE can outperform CoDA and GEPS with few-shot test-time adaptation, due to their difference on discovering PDE invariance. Specifically, iMOOE explicitly prescribes the two-level PDE invariance principle and effectively approximates it via the proposed physics-informed mixture of operator expert architecture and invariant learning objective. While meta-learning-based CoDA, GEPS assume PDE invariance lies in domain-invariant network parameters. They implicitly learn domain-generalizable representations in the parameter space without any physical guidance.

Table 19: Further comparisons with meta-learning-based methods.

| Methods | BG | | NS | |
|---|---|---|---|---|
| | nMSE | fRMSE | nMSE | fRMSE |
| CoDA(zero-shot) | 9.22e-1 | 2.50e-2 | 9.14e-1 | 7.31e-2 |
| CoDA(few-shot) | 6.89e-1 | 1.84e-2 | 6.31e-1 | 6.45e-2 |
| GEPS(zero-shot) | 7.56e-2 | 9.38e-3 | 4.13e-1 | 6.85e-2 |
| GEPS(few-shot) | 5.37e-2 | 6.58e-3 | 3.32e-1 | 5.47e-2 |
| iMOOE(zero-shot) | **1.08e-2** | **3.83e-3** | **3.12e-1** | **5.36e-2** |

### F.5 EMPIRICAL UPPER BOUND OF ZERO-SHOT OOD PERFORMANCE

To more intuitively gauge iMOOE's zero-shot OOD forecasting capability, we propose to verify the empirical upper bound for iMOOE's OOD performance. Such bound can measure iMOOE's achievable OOD performance on unseen domains with distribution shifts (Gagnon-Audet et al., 2023). Akin to the test operation in (Gagnon-Audet et al., 2023), we randomly select four OOD test domains in DR data and train the standard FNO under each specific environment with different volumes of training trajectories. In Table 20, we compare the OOD performance of zero-shot iMOOE with three levels of empirical upper bounds forged by FNO. Apparently, iMOOE can *consistently surpass the 16-shot FNO* and *rival the 64-shot FNO*, while underperforming the 256-shot FNO. It further demonstrates the proposed PDE invariance learning can improve the zero-shot OOD capability of neural operators on unseen scenarios.

Table 20: Empirical upper bound of zero-shot OOD capacity on DR data.

| Models | Env1 | | Env2 | | Env3 | | Env4 | |
|---|---|---|---|---|---|---|---|---|
| | nMSE | fRMSE | nMSE | fRMSE | nMSE | fRMSE | nMSE | fRMSE |
| FNO(256-shot) | **3.40e-3** | **8.90e-4** | **3.65e-3** | **6.56e-4** | **1.38e-3** | **4.09e-4** | **4.59e-3** | **7.69e-4** |
| FNO(64-shot) | 6.38e-2 | 3.71e-3 | 6.20e-2 | 1.95e-3 | 1.18e-2 | 1.23e-3 | 4.34e-2 | 2.36e-3 |
| FNO(16-shot) | 2.10e-1 | 6.71e-3 | 9.41e-2 | 3.30e-3 | 4.06e-2 | 2.08e-3 | 1.27e-1 | 3.95e-3 |
| FNO-iMOOE(zero-shot) | 1.27e-2 | 1.65e-3 | 5.12e-2 | 1.82e-3 | 6.10e-2 | 1.42e-3 | 4.30e-2 | 2.09e-3 |

### F.6    More Results on ID-OOD Correlations in Fig. 1(c)

In the scope of domain generalization, ID-OOD correlation (Miller et al., 2021; Yuan et al., 2023) is a useful metric to reflect the effective OOD robustness of a deep learning model. If the relationships between ID and OOD test errors are sharply positive (i.e. the slope of ID-OOD fitted line is positively large), we can claim that the developed neural network indeed captures the domain-generalizable representations from training data and its OOD robustness is satisfactory. In practice, the ID-OOD correlation line can be obtained by testing the developed model under various training hyper-parameters, such as changing the quantity of training data, total epochs, initial learning rates, etc. For example, a single blue scatter in Fig. 5 represents the FNO-iMOOE model with a unique training configuration. The same interpretations for the orange scatter of FNO. As the slope of FNO-iMOOE's ID-OOD line is significantly sharper than that of FNO, we can state that when FNO is augmented by the proposed PDE invariance learning framework, it is able to capture the fundamental invariance in PDE dynamics and achieve better OOD forecasting performance.

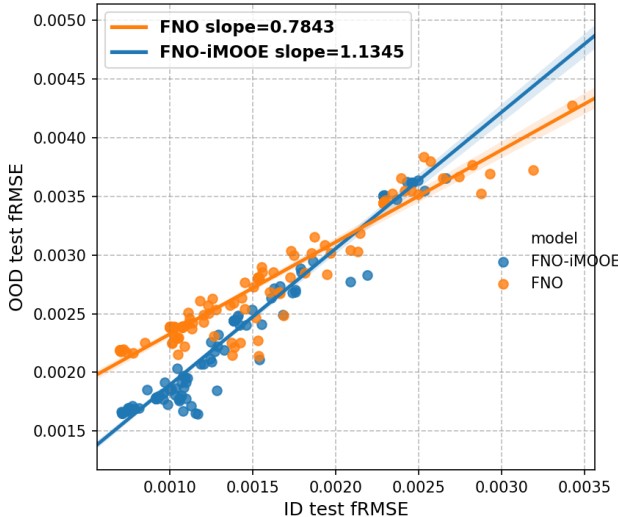

Figure 5: Supplementary fRMSE results for ID-OOD correlations.

### F.7    Analysis on Training Data Properties

In practice, either measuring real-world dynamics trajectories by multi-source sensors or generating simulated PDE data by numerical solvers is prohibitively expensive. To this end, it is of great significance to investigate the impact of training data properties on zero-shot OOD forecasting capability. This can guide us to construct more informative multi-context sequences and further improve OOD performance from the data perspective. We conduct this study by answering two questions: i) What is the effect of training data quantity? ii) When the budget of collecting training data is limited, in terms of data diversity (i.e. the number of training environments $|\mathcal{E}_{tr}|$) and data quantity within each environment, which factor is more important? DR data is utilized to probe these two aspects of data properties. We showcase corresponding fRMSE and nMSE results in Fig. 6 and Fig. 7.

For the first question, we escalate the size of training trajectories from 256 to 4,096. Overall, with the size of training data increasing, ID/OOD generalization capacity of PDE forecasting models elevate considerably, which is amenable to the scaling property between data size and model performance in scientific machine learning (Subramanian et al., 2023). Notably, for OOD fRMSE results, iMOOE trained on size 512 can rival FNO trained on size 2,048. For OOD nMSE results, iMOOE trained on size 512 even outperforms FNO trained on size 4,096. It indicates that the proposed PDE invariance learning can enhance the zero-shot OOD performance and data efficiency of ordinary FNO. Using 1,024 training trajectories for iMOOE can reach satisfactory zero-shot OOD results on DR dynamics compared to naive FNO.

For the second question, we keep the total number of training samples at 1,024 and alter the number of training environments from 4 to 512. The training data quantity in each environment is equal.

We depict the distribution of ID and OOD results of each test sample in Fig. 6(b), 7(b). Overall, with diverse training domains, i.e. when the number of training environments is up to 32, ID/OOD results of each test trajectory can disperse more compactly. In other words, the variance across test domains is smaller, and the average ID/OOD fRMSE is much lower. This reveals that when data budget is limited, better data diversity can avoid overfitting to limited training domains, and aid to find the fundamental PDE invariance principle by equalizing the risks across more diverse training environments. This is coherent with the key claim in foundational invariant learning literature (Arjovsky et al., 2019): with a sufficiently large number of diverse training environments, invariant risk minimization would elicit the invariant predictor.

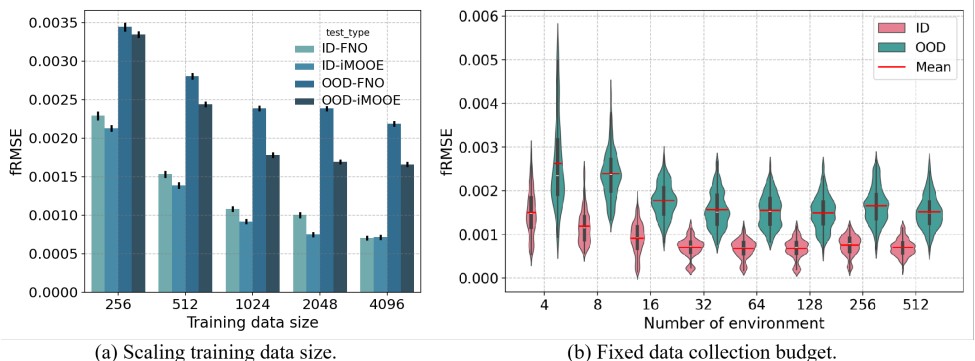

(a) Scaling training data size.         (b) Fixed data collection budget.

Figure 6: Impact of training data properties on ID/OOD fRMSE from two views: (a) Varying data size. (b) Varying data diversity under limited data budget.

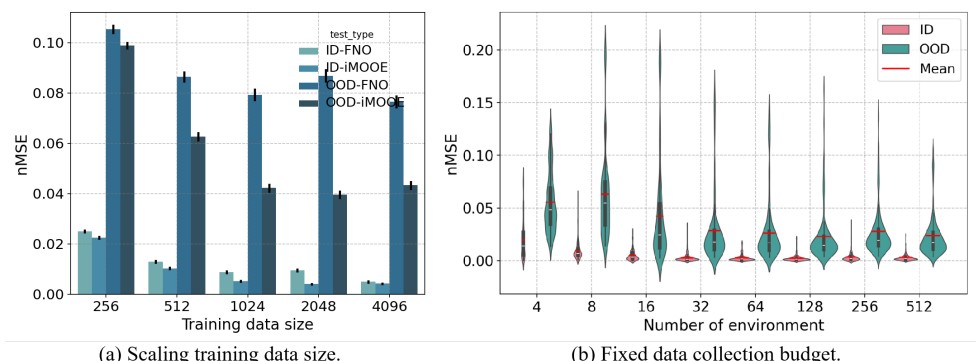

(a) Scaling training data size.         (b) Fixed data collection budget.

Figure 7: Impact of training data properties on ID/OOD nMSE from two views: (a) Varying data size. (b) Varying data diversity under limited data budget.

### F.8 RUNTIME COMPARISON

We compare the runtime of both the PDE forecasting methods presented in Table 1 and the commercial numerical solver Comsol (Multiphysics, 1998) in Table 21. We can see that the deep learning methods can lead to a nearly $225 \times$ times speed-up on inferring the BG flow trajectories in contrast to the inner finite element method in Comsol. It is hard for Comsol to converge when simulating the turbulent flow (i.e. the viscosity coefficient $\nu$ in BG is small). At the same time, neural PDE methods can obviate the need for complicated domain knowledge on modeling the real-world PDE systems. Besides, it is apparent that FNO-iMOOE indeed incurs extra computational burden on top of vanilla FNO, while its running speed is similar to other OOD forecasting methods for PDE dynamics.

Table 21: Runtime comparison of different PDE dynamics simulation methods on BG data.

| Methods | Comsol | FNO-iMOOE | FNO | CAPE | VCNeF | DPOT | CNO | GEPS | CoDA |
|---|---|---|---|---|---|---|---|---|---|
| Inference Time | 24.84±2.73s | 0.11±0.002s | 0.05±0.002s | 0.06±0.002s | 0.11±0.003s | 0.09±0.002s | 0.12±0.004s | 0.11±0.003s | 0.07±0.002s |

### F.9 IMPLEMENTATION DETAILS

We clarify hyperparameter settings for all baseline methods in Table 1, 2. The fixed training setups include 32 training batch size and $1e-3$ initial rate for Adam optimizer.

- **CoDA.** We train CoDA for 1500 epochs. The hidden dimension of shared hypernetwork and domain-specific 4-layer FNOs is 64. The weight of its $L_1$ and $L_2$ regularization on hypernetwork parameters is $1e-5$.
- **CAPE.** We train CAPE for 500 epochs. The widdening factor of its channel attention and width of 4-layer FNO backbone are 64. The weight of additional loss $\mathcal{L}_{cape}$ is $8.3e-5$.
- **CNO.** We train CNO for 500 epochs. The channel multiplier of its UNet-shaped operator is 16. The hidden dimension and layer number of its bottleneck network is 128 and 4.
- **DPOT.** We finetune the pretrained DPOT of tiny version for 500 epochs. The latent dimension of Fourier attention and FFN layer is 512. The number of attention head is 4.
- **VCNeF.** We train VCNeF for 500 epochs. The latent dimension and patch size of the linear transformer block is 64 and 16. The depth of modulation blocks is 4.
- **GEPS.** We train GEPS for 1500 epochs. The width of domain-specific 4-layer FNO is 64 and code size of context vector is 16.

### F.10 VISUALIZATION ON OOD FORECASTING RESULTS OF iMOOE

In Fig. 8 to 14, we visualize the forecasting outcomes of iMOOE on representative OOD physical environments of the five PDE dynamical systems.

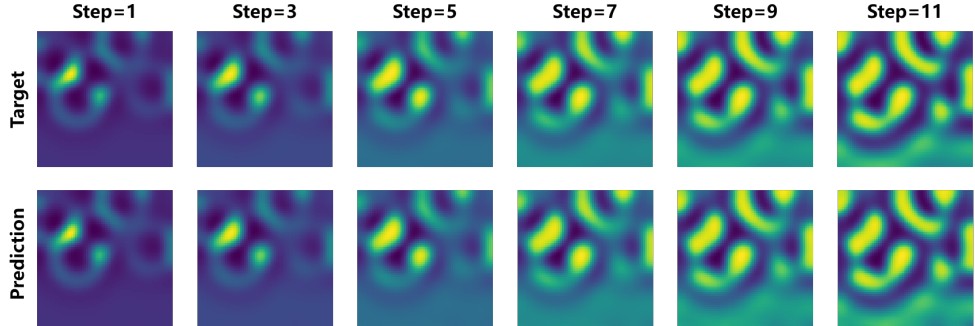

Figure 8: OOD forecast showcase on a DR scenario with $D_u = 0.0021, D_v = 0.0113, k = 0.0109$.

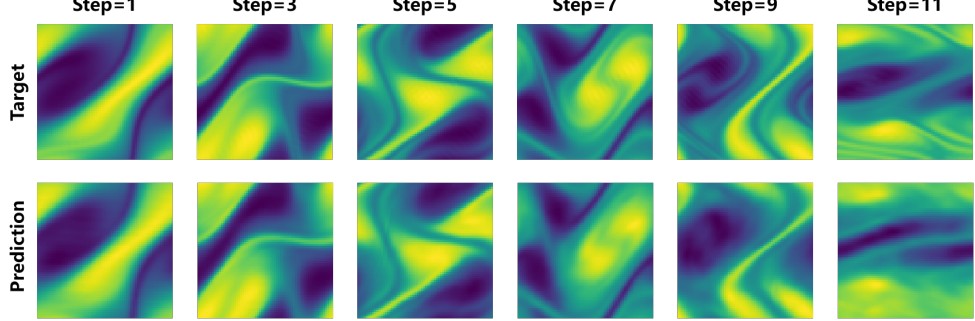

Figure 9: OOD forecast showcase on a high-Reynold number NS scenario with $\nu = 1.42e\text{-}4$.

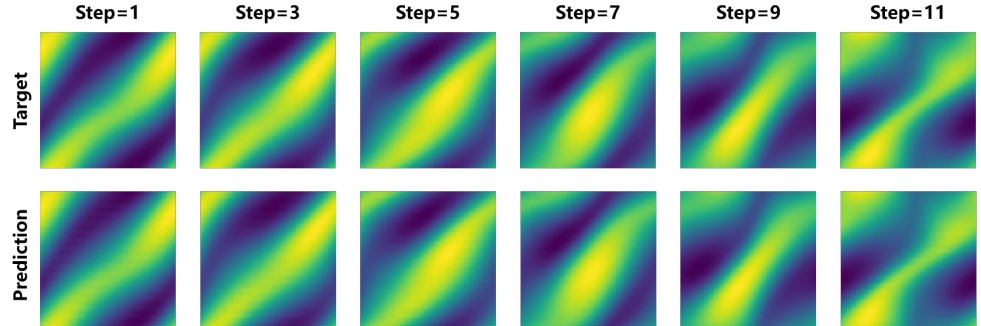

Figure 10: OOD forecast showcase on a low-Reynold number NS scenario with $\nu = 1.20e$-3.

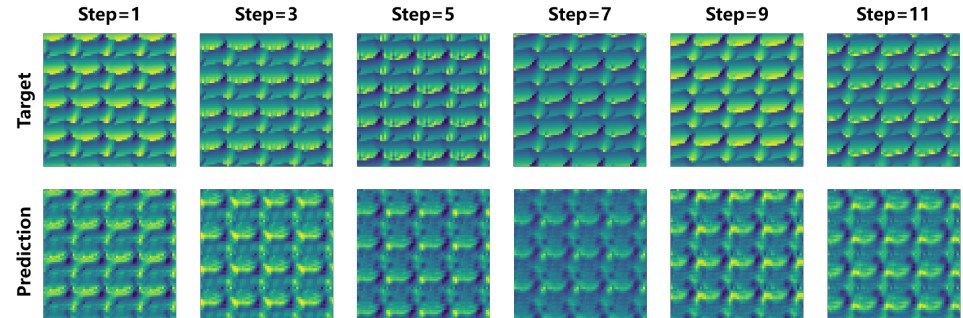

Figure 11: OOD forecast showcase on a high-Reynold number BG scenario with $\nu = 2.5e$-3.

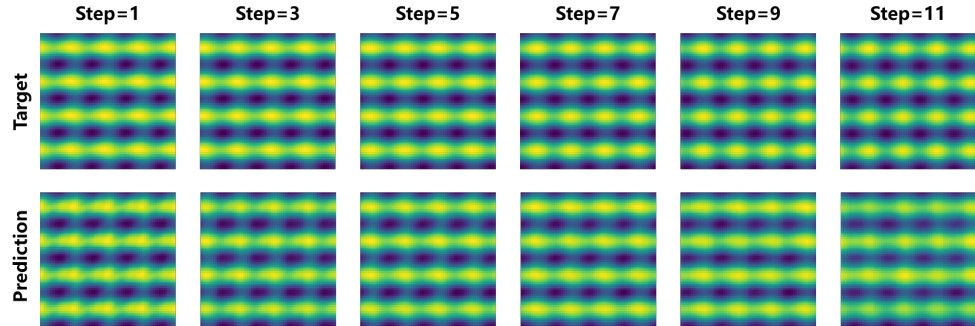

Figure 12: OOD forecast showcase on a low-Reynold number BG scenario with $\nu = 1.0e$-1.

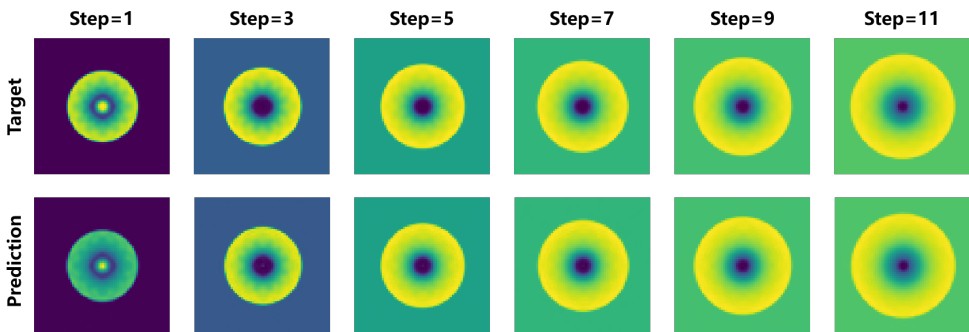

Figure 13: OOD forecast showcase on a SW scenario with an unseen initial radius.

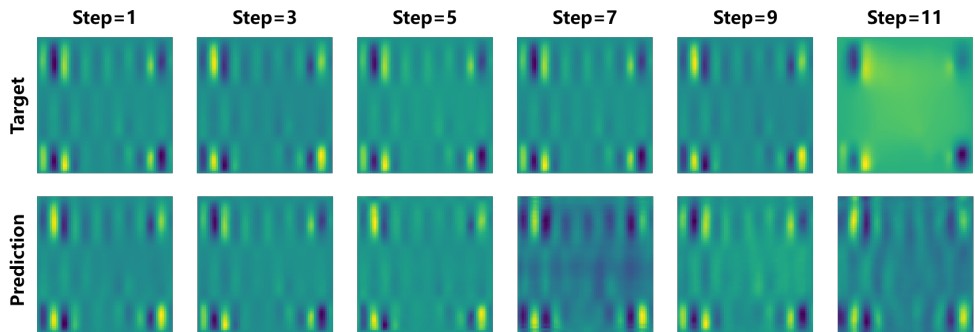

Figure 14: OOD forecast showcase on a HC scenario with $m_1 = 2.67, m_2 = 12.66, m_3 = 2.74$.

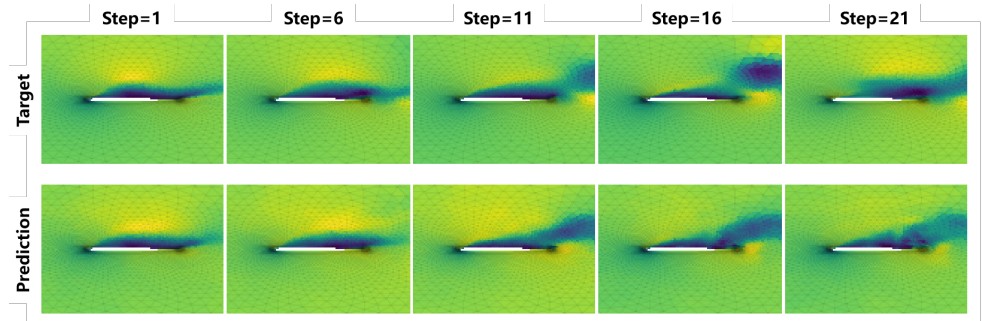

Figure 15: OOD forecast showcase on the x-axis velocity around Airfoil with unseen conditions.

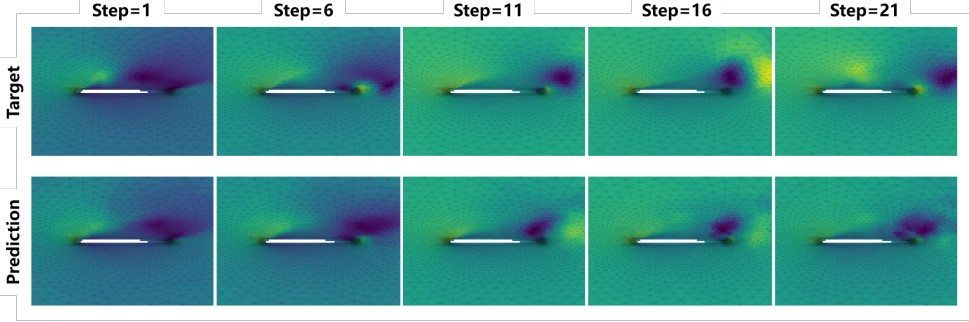

Figure 16: OOD forecast showcase on the y-axis velocity around Airfoil with unseen conditions.

