# OpenReview forum: "Towards Generalizable PDE Dynamics Forecasting via Physics-Guided Invariant Learning"
_ICLR.cc/2026/Conference — ICLR 2026 Poster_

### Official Review · Reviewer_YiSh · 2025-10-22

**Soundness:** 3
**Presentation:** 3
**Contribution:** 3
**Rating:** 6
**Confidence:** 3

**Summary:**

This paper tackles zero-shot OOD generalization for PDE forecasting with **iMOOE**, a framework built on a **"two-fold PDE invariance principle"**. This principle posits that core physical operators and their compositional structure remain invariant despite parameter shifts. The framework captures this using an **invariance-aligned Mixture of Operator Expert (MOOE) architecture**  and a **frequency-enriched invariant learning objective**. This objective enforces risk equality while also using a novel regularizer to **mitigate the spectral bias** of neural operators, forcing them to learn essential high-frequency features. Experiments show this "plug-and-play" framework significantly boosts zero-shot OOD performance for various neural operators.

**Strengths:**

* The paper tackles the highly significant and challenging problem of zero-shot OOD generalization for PDE dynamics. It accurately pinpoints the failure of prior methods: they do not capture the underlying physical invariance required for true zero-shot generalization without test-time adaptation.

* The core originality lies in proposing a "two-fold PDE invariance principle" (operator and compositionality invariance). This provides a structured, physics-based hypothesis that reframes the problem from finding abstract features to learning specific, invariant physical components.

* The framework is technically novel and significant in two ways. First, its "plug-and-play" nature makes it a broadly applicable tool for enhancing various existing neural operators. Second, it originally introduces a **frequency-enrichment loss** ($\mathcal{L}_{freq}$) to mitigate the spectral bias of neural operators, which is a critical and novel step for learning the complete physical invariance.

* The paper's claims are supported by comprehensive experiments across simulated PDEs and a real-world dataset, using a strict OOD setup with non-overlapping parameters. The results consistently show SOTA performance. Crucially, the insightful ablation studies (e.g., on expert number $K$ and the $\mathcal{L}_{freq}$ loss) effectively validate that each core component of the framework contributes to the final success.

**Weaknesses:**

* The paper's central premise—that prior methods fail *because* they miss the "fundamental invariance principle"—is asserted without direct evidence. This creates a correlation-not-causation gap, as the paper only proves its own method works, not *why* others fail.

* The paper fails to validate its core hypothesis with interpretability. There is no analysis or visualization of *what* the specialized experts ($\sigma_i$) actually learn, nor any theoretical analysis of the decomposition's properties (e.g., completeness, uniqueness).

* The mechanism for learning the derivative selection masks ($m_i$) is ambiguous. It is unclear whether these masks are learnable parameters or the output of a separate network, which harms reproducibility.

* A crucial baseline is missing: a monolithic neural operator with an equivalent parameter count to the final $K$-expert MOOE model. Without this control, it is unclear if the performance gain comes from the principled MOOE *structure* or simply from increased model capacity.

**Questions:**

1.  The paper's premise is that prior OOD methods fail *because* they miss the "fundamental invariance principle." Is there direct evidence to support this causal claim about *why* baselines fail, beyond iMOOE's own success?

2.  To validate the core hypothesis, can you provide an interpretability analysis of *what* the individual experts $\sigma_i$? For example, in the DR system, do they measurably specialize into distinct diffusion and reaction operators?

3.  Could you clarify the exact implementation of the derivative selection masks $m_i$? Are they learnable parameter vectors for each expert, or the output of a separate routing/gating network?

4.  To isolate the benefit of the MOOE *structure* from simply having more parameters, could you add a baseline comparison against a single, monolithic operator with an equivalent total parameter count to the $K$-expert iMOOE?

5.  While iMOOE uses dense experts, does it face training challenges analogous to sparse LLM MoEs, such as expert collapse or redundancy (especially as $K$ increases)? How effective is $\mathcal{L}_{mask}$ at ensuring true specialization versus mere diversity?

---

> ### Author Response · Authors · 2025-11-20
> **Response to Reviewer YiSh (Part 1)**
>
> *We greatly appreciate your positive reviews regarding iMOOE. Hope our response can address your questions and concerns.*
>
> **W1\&Q1: Is there direct evidence to support why baselines fail?**
>
> Previous baselines like meta-learning-based CoDA, GPES or parameter conditioning-based CAPE just assume: domain-invariant network parameters which are not perturbed by varying contexts can capture the invariance in PDE dynamics. However, they do not meet the requirement of invariant learning from two aspects: i) Training objective. As stated in Assumption 2.1 of [2], the invariant representations should satisfy both invariance property and sufficiency property. Baseline methods just employ the predictive loss to minimize the forecasting error, which only meet the sufficiency property. In contrast, iMOOE additionally employ the risk equality loss to meet the invariance property. ii) Invariance principle. Baseline methods do not explicitly elucidate the invariance principle in PDE dynamics forecasting, thus their architecture design might not be suitable to capture PDE invariant representations [2]. In contrast, iMMOE is aligned with the clarified two-level PDE invariance. Experimental results validate it can perform better on capturing invariant representations in PDE trajectories and zero-shot OOD generalization.
>
> **W2\&Q2: No interpretability analysis on what the specialized experts actually learn.**
>
> For simplicity, we analyze the output sequences of two FNO experts on the linear Diffusion-Reaction PDE forecasting. We can calculate the Structure Similarity Index Measure (SSIM) between each expert output sequence and the distinct Diffusion or Reaction process trajectory. SSIM=1 indicates two images are identical. By computing SSIM on 128 OOD DR samples, we observe that the average SSIM between one FNO output sequence and the Reaction dynamics trajectory is 0.7750. The average SSIM between another FNO output sequence and the Diffusion dynamics trajectory is 0.6805. Such similarity results indicate that two FNO experts of iMOOE can approximate the distinct Diffusion and Reaction operators respectively. We showcase two examples of FNO expert output from a random OOD environment in **Figure 17, 18 of revised paper**.
>
> Actually, there are multiple ways to decompose a PDE into separate operators. iMOOE will learn its own operator splitting way via the proposed physics-guided invariant learning from diverse training environments. It is quite challenging to interpret what the individual operator expert really represent, especially for the complex nonlinear PDE systems like Navier-Stokes. We will leave this operator representation analysis problem for the future work.
>
> **W3\&Q3: The mechanism for learning the derivative selection masks is ambiguous.**
>
> The derivative selection mask $\mathbf{m} _{i}$ for each operator expert is a learnable binary vector which is implemented by nn.Parameter() in PyTorch and supports gradient backpropagation. We initialize $\mathbf{m} _{i}$ using the standard normal distribution. The elements in $\mathbf{m} _{i}$ are mapped to 1 (indicating the corresponding pre-calculated derivative should be selected as input) if they are larger than 0, otherwise they are mapped to 0 (indicating the corresponding pre-calculated derivative should be masked out). We utilize the Straight Through Estimator (STE) in PyTorch to enable gradient backpropagation for $\mathbf{m} _{i}$ optimization, which involves mapping continuous values to discrete 0-1. Please refer to the supplementary iMOOE code package, where class STEFunction(torch.autograd.Function) shown in "models/framework.py" provides the clear implementation of the proposed mask mechanism.

---

> ### Author Response · Authors · 2025-11-20
> **Response to Reviewer YiSh (Part 2)**
>
> **W4\&Q4: Isolate the benefit of MOOE structure by adding a monolithic operator baseline with equivalent model capacity.**
>
> According to Appendix B.2 in [1], enlarging FNO size by increasing the hyperparameter "width" can improve FNO's forecasting accuracy on PDE trajectories. To validate the benefit of physics-guided MOOE architecture design, we expand the width of FNO to render FNO own a model capacity closest to FNO-MOOE. In the following table, we find that FNO-MOOE aligning with the proposed PDE invariance principle can outperform brute-force scaling up the model size of FNO.
>
> | Models | NS |  |  |  | SST |  |  |  |
> | :---: | :---: | :---: | :---: | :---: | :---: | :---: | :---: | :---: |
> |  | Width | Params | nMSE | fRMSE | Width | Params | nMSE | fRMSE |
> | FNO | 94 | 40,860,767 | 3.63e-1 | 5.87e-2 | 52 | 8,366,194 | 6.37e-1 | 4.01e-2 |
> | FNO-MOOE | 64 | 40,547,901 | **3.21e-1** | **5.40e-2** | 32 | 8,441,097 | **5.82e-1** | **3.71e-2** |
> | Improvement |  |  | 11.57\% | 8.01\% |  |  | 10.20\% | 7.48\% |
>
>
> **Q5: Does iMOOE face expert collapse or redundancy, and ensure true specialization?**
>
> The expert collapse issue does not occur in iMOOE training process since we do not develop a gating network to select a subset of operator experts. As many PDE systems are composed of a few number of distinct operators, there may exist representation redundancy when integrating an increasing number of operator experts into iMOOE. In **Table 6 of revised paper**, we observe that iMOOE's OOD performance can degrade when K=4. We also observe the overlapped input derivative selection across 4 experts in **Appendix D.2 of revised paper**.  According to **Table 6 of revised paper**, K=2 and K=3 can notably outperform k=1, indicating the additional operator experts are specialized in approximating the distinct sub-operators. In **Appendix E.1.2 and Table 9 of revised paper**, we demonstrate the proposed derivative masking scheme with diversity loss can notably improve iMOOE's zero-shot OOD results on different PDE systems. It indicates the mask diversity loss can help iMOOE to learn the operator invariance.
>
>
> [1] Incremental Spatial and Spectral Learning of Neural Operators for Solving Large-Scale PDEs, Transactions on Machine Learning Research, 2024.
>
> [2] Heterogeneous Risk Minimization, ICML 2021.

---

> > ### Comment · Reviewer_YiSh · 2025-11-22
> >
> > Thank you for the additional experiments and clarifications. However, I remain unconvinced regarding the response to Q2. While the authors provide SSIM analysis on the DR system, the evidence for 'operator specialization' remains inconclusive. Specifically, the reported SSIM of 0.68 for the Diffusion expert is relatively low, suggesting incomplete disentanglement. More critically, the authors omit the cross-correlation metrics (e.g., SSIM between the Reaction Expert and the Diffusion trajectory). Without comparing diagonal vs. off-diagonal terms in the confusion matrix, we cannot verify if the experts are truly specialized or simply learning redundant, mixed representations. The claim of physics-guided decomposition is empirically unsupported, even if the final predictive performance is strong.

---

> > > ### Author Response · Authors · 2025-11-24
> > > **Reponse to Reviewer YiSh (Further Question)**
> > >
> > > We greatly appreciate for your meticulous review on our response as well as the further insightful question. To verify operator experts can learn disentangled PDE representations and capture distinct physical processes, we can directly calculate SSIM between the output trajectories of two independent experts. We can decode one expert output by masking out the other expert during the forward pass. We experiment on two simulating and one real-world PDE datasets, and report the minimum, maximum and mean SSIM over OOD test samples in the following table. The hyperparameter configurations of iMOOE are identical to those in Table 1-3 of revised paper. We can observe that the output trajectories of two operator experts on three datasets are dissimilar to each other. Hence, two FNO experts of iMOOE can be specialized in distinct physical processes, instead of tending to learn redundant PDE representations. The split operator architecture design and mask diversity loss can help iMOOE to learn the distinct invariant operators.
> > >
> > > | Dataset | min SSIM | max SSIM | mean SSIM |
> > > | :---: | :---: | :---: | :---: |
> > > | DR | 0.0014 | 0.2458 | 0.1235 |
> > > | NS | 0.0672 | 0.4388 | 0.2301 |
> > > | SST | 0.2405 | 0.4992 | 0.3698 |

---

> ### Comment · Reviewer_YiSh · 2025-11-27
>
> Thanks, I will keep my score towards acceptance.

---

> > ### Author Response · Authors · 2025-11-28
> > **Response to Reviewer YiSh**
> >
> > We sincerely appreciate for your positive support and insightful suggestions for our work. Thanks a lot!

---

### Official Review · Reviewer_K9mK · 2025-10-27

**Soundness:** 2
**Presentation:** 3
**Contribution:** 3
**Rating:** 4
**Confidence:** 3

**Summary:**

This paper proposes iMOOE, a physics-guided invariant learning framework for zero-shot out-of-distribution (OOD) generalization in PDE dynamics forecasting. The key idea is to exploit two levels of PDE invariance—operator invariance and compositional invariance—through a Mixture of Operator Experts (MOOE) architecture and a frequency-enriched invariant learning objective. The authors show improved ID and OOD performance across several simulated PDE systems (Diffusion–Reaction, Navier–Stokes, Burgers, Shallow-Water, and Heat-Conduction) compared with recent meta-learning and parameter-conditioned models.

**Strengths:**

1. The paper tackles an important and difficult problem “zero-shot OOD generalization in PDE dynamics” and does so with a physically motivated approach.

2. The proposed Mixture of Operator Experts (MOOE) architecture is intuitive and nicely aligned with operator-splitting ideas in classical PDE solvers.

3. The frequency-augmented loss and risk-equality objective are interesting design choices that improve empirical robustness.

4. Experiments on multiple PDE benchmarks show clear performance gains over meta-learning and conditioning-based methods.

**Weaknesses:**

1. The “two-level PDE invariance” principle feels more heuristic than theoretically grounded — there’s no rigorous justification or proof that these invariances hold across PDE families.

2. All experiments are on synthetic, noise-free datasets with simple geometries and periodic boundaries. No real or noisy data are tested, so generalization claims are not fully convincing.

3. The frequency-based regularization might amplify noise in realistic cases, but this is not examined.

4. Comparisons with baselines may not be entirely fair since meta-learning models are designed for few-shot adaptation rather than strict zero-shot settings.

**Questions:**

1. Can the authors provide more formal or causal justification for the “two-level invariance” concept?

2. How sensitive is the method to the way training environments are defined (by parameters vs. time steps)?

3. How does the model perform on noisy or irregular real-world PDE data, such as CFD or weather simulations?

4. Does the frequency loss harm stability when data are non-periodic or contain sensor noise?

5. What is the computational overhead of iMOOE compared to a standard FNO?

---

> ### Author Response · Authors · 2025-11-20
> **Response to K9mK (Part 1)**
>
> *Many thanks for your meticulous review and insightful questions. In the following, we make great effort to address each of your concerns and questions.*
>
> **W1\&Q1: Provide more rigorous justification for the two-level PDE invariance principle.**
>
> Please refer to **W2-1 response to Reviewer adhf** for the rigorous justification of proposed operator invariance and compositionality invariance.
>
> **W2\&Q3: Lacking generalization results on real-world noisy or irregular PDE data.**
>
> In **Section 4.4 of original manuscript**, we have leveraged a real-world Ocean-Atmosphere dynamics dataset called Sea Surface Temperature (SST, PDE-governed upper-ocean thermodynamics) to validate iMOOE's zero-shot generalization performance. SST dataset contains real sensor noise which originates from satellite radiometers, ship measurements, buoys and Argo floats [3]. **Table 4 and Figure 3 of original manuscript** have demonstrated iMOOE can better generalize to unseen physical contexts and capture local SST variations with higher fidelity.
>
> Apart from SST dataset, we additionally utilize another real-world Ocean-Atmosphere dynamics dataset called stereo Sea Surface Elevation (SSE, PDE-governed free-surface ocean wave dynamics) [4] to verify iMOOE's zero-shot generalization capability. SSE dataset contains real measurement noise due to the raw stereo images and wave field reconstruction, including stereo-matching errors, camera calibration uncertainty and environmental visual noise. We choose the wave dynamics recorded at La Jument lighthouse with total 4500 frames on $241\times 221$ grid. The training and test set are formed by the first 4000 frames and remaining 500 frames. Following SST implementation, we input past 4 steps SSE state to predict future 6 steps. In the table below, we demonstrate iMOOE can achieve the best OOD generalization performance on SSE dynamics compared to other models. We have included such SSE results into **Section 4.4, Table 5 and Figure 4 of the revised paper**. Regarding PDE forecasting on irregular grids, please refer to **Q1 response to Reviewer RXdg**.
>
> | Models | nMSE |  | fRMSE |  |
> | :---: | :---: | :---: | :---: | :---: |
> |  | Mean | Std | Mean | Std |
> | GEPS | 3.41e-2 | 4.96e-3 | 4.11e-3 | 1.55e-4 |
> | DPOT | \ul 2.46e-2 | \ul 3.28e-3 | \ul 3.54e-3 | \ul 1.49e-4 |
> | VCNeF | 3.41e-2 | 4.96e-3 | 4.11e-3 | 1.55e-4 |
> | DyAd | 3.17e-2 | 4.61e-3 | 3.96e-3 | 1.50e-4 |
> | CAPE | 3.34e-2 | 4.85e-3 | 4.07e-3 | 1.53e-4 |
> | **iMOOE** | **1.52e-2** | **2.34e-3** | **2.78e-3** | **1.42e-4** |
>
>
> **W3\&Q4: Frequency-based regularization might amplify noise in real-world data.**
>
> In real-world PDE-governed dynamics data with measurement noise such as SST [3] and SSE [4], the real high-frequency patterns still exist in their spatial domain and should be attended for better generalization. For example, SSE contains high-frequency short waves which could be caused by the nonlinear surface features like wave breaking fronts, sharp crests and bound harmonics. SST contains high-frequency modes due to the ocean advection and vertical processes such as upwelling. In the following table, we present that the proposed frequency enrichment loss can improve nMSE by $9.22\%$ and $8.98\%$, fRMSE by $7.03\%$ and $7.33\%$ on two datasets. It is obvious that regularizing high-frequency feature learning can also enhance the OOD accuracy on low-frequency and mid-frequency patterns. We have included these results in **Table 8 in Appendix E.1.1 of revised paper**.
>
> | Methods | SST |  |  |  |  | SSE |  |  |  |  |
> | :---: | :---: | :---: | :---: | :---: | :---: | :---: | :---: | :---: | :---: | :---: |
> |  | nMSE | fRMSE |  |  |  | nMSE | fRMSE |  |  |  |
> |  |  | Low | Mid | High | Total |  | Low | Mid | High | Total |
> | w/o $\mathcal{L}_{freq}$ | 5.64e-1 | 1.29e-1 | 3.55e-2 | 9.24e-3 | 3.70e-2 | 1.67e-2 | 3.03e-3 | 2.98e-3 | 3.01e-3 | 3.00e-3 |
> | w/ $\mathcal{L}_{freq}$ | **5.12e-1** | **1.24e-1** | **3.06e-2** | **8.47e-3** | **3.44e-2** | **1.52e-2** | **2.10e-3** | **2.74e-3** | **2.94e-3** | **2.78e-3** |
>
>
> **W4: Comparisons with meta-learning-based models may not be fair.**
>
> In **Appendix E.4 and Table 19 of original manuscript**, we have demonstrated our proposed zero-shot iMOOE can outperform few-shot CoDA, GEPS, due to iMOOE's explicit learning on physics-guided PDE invariance principle. We clarified that meta-learning-based methods can be adapted for zero-shot OOD test, by initializing domain-specific network parameters (i.e. $\theta^{e}$ in CoDA, $\mathbf{c}^{e}$ in GEPS) using averaged parameters optimized over diverse training domains. We also comply to the standard few-shot implementation claimed in Section 5.2 of native CoDA [1] and GEPS [2] papers, which draws one trajectory from each unseen OOD environment for test-time adaptation.

---

> ### Author Response · Authors · 2025-11-20
> **Response to K9mK (Part 2)**
>
> **Q2: Sensitivity to two ways of training environment partition.**
>
> In the following table, we validate iMOOE's sensitivity to different environment partition methods on two fluid dynamics forecasting tasks. Apparently, combining two partition methods together can realize the best zero-shot OOD performance, since it can enhance the diversity of training environments and improve the robustness of learned PDE invariance representations. Besides, parameter-based partition performs moderately better than step-wise partition, as different physical parameters can lead to more distinct PDE trajectories, such as the Reynold number in NS is a decisive factor to distinguish laminar or turbulent flow. We have appended these results and discussions to **Appendix E.1.4 and Table 11 of revised paper**.
>
> | Partition methods | NS |  | BG |  |
> | :---: | :---: | :---: | :---: | :---: |
> |  | nMSE | fRMSE | nMSE | fRMSE |
> | Only parameters | 3.41e-1 | 5.49e-2 | 1.11e-2 | 3.94e-3 |
> | Only time steps | 3.58e-1 | 5.52e-2 | 1.18e-2 | 4.01e-3 |
> | Parameters+time steps | **3.12e-1** | **5.36e-2** | **1.08e-2** | **3.83e-3** |
>
>
> **Q5: iMOOE's computational overhead compared to standard FNO.**
>
> The introduction of additional FNO operator experts can induce the increase of computation costs compared to standard FNO, as shown in **Table 6 of revised paper**. And in **Table 21 of revised paper**, we have compared iMOOE's inference time with FNO and other baseline models. For a PDE trajectory with 10 steps input and 11 steps output on $64\times 64$ grid, the inference time of FNO and iMOOE with two FNO experts is 0.05s and 0.11s. We can reduce the computational overhead by integrating more computationally efficient neural operators into iMOOE.
>
>
> [1] Generalizing to New Physical Systems via Context-Informed Dynamics Model, ICML 2022.
>
> [2] Boosting Generalization in Parametric PDE Neural Solvers through Adaptive Conditioning, NeurIPS 2024.
>
> [3] Improvements of the daily optimum interpolation sea surface temperature (DOISST) version 2.1, Journal of Climate, 2021.
>
> [4] A data set of sea surface stereo images to resolve space-time wave fields, Scientific Data, 2020.

---

> ### Author Response · Authors · 2025-11-27
> **Kind Request from Authors**
>
> Dear Reviewer K9mK,
>
> We hope this message finds you well. As the discussion period is approaching to the final stage, could we kindly ask whether our responses have addressed your concerns and questions? In our rebuttal, we provided more rigorous justification for the proposed two-level PDE invariance, as well as additional experimental results to further verify the effectiveness of iMOOE design. We would greatly appreciate your thoughts and further comments or suggestions on our revised work. Thank you for your time and consideration.

---

### Official Review · Reviewer_adhf · 2025-10-27

**Soundness:** 3
**Presentation:** 3
**Contribution:** 3
**Rating:** 6
**Confidence:** 2

**Summary:**

This paper addresses the challenge of out-of-distribution (OOD) generalization in PDE dynamics forecasting. The paper proposes iMOOE, consisting of:
- A Mixture of Operator Experts (MOOE) architecture, where parallel neural operator experts model distinct physical processes, and a fusion network captures compositional relationships.
- A frequency-enriched invariant learning objective, combining prediction, risk equality, and frequency regularization losses to promote invariant representations across environments.
Extensive experiments on five PDE systems (Diffusion-Reaction, Navier–Stokes, Burgers, Shallow-Water, Heat-Conduction) show that iMOOE improves ID and OOD generalization over baselines.

**Strengths:**

- Architectural Innovation:
The MOOE architecture aligns with PDE operator decomposition, offering modularity and interpretability. Its parallel design is more efficient than serial operator splitting.

- Robust Empirical Validation:
The experiments are comprehensive—covering both synthetic PDE benchmarks and a real-world application (SST).

**Weaknesses:**

- Limited Diversity of PDE Systems:
Although the paper evaluates multiple canonical PDEs, they are all relatively low-dimensional and well-behaved (2D systems with periodic boundaries). The scalability to irregular geometries or 3D turbulent systems is unclear.

- Theoretical Rigor of Invariance Definition:
The proposed “two-level PDE invariance” is intuitive but lacks rigorous formal grounding. The connection between the invariance principle and the optimization objective could be better substantiated.

**Questions:**

See weaknesses

---

> ### Author Response · Authors · 2025-11-20
> **Response to Reviewer adhf (Part 1)**
>
> *We are grateful for your positive support and beneficial comments. Below we address each question and concern.*
>
> **W1: Limited diversity of PDEs: scalability to irregular geometries or 3D turbulent systems is unclear.**
>
> One of the key advantages of iMOOE is its elegant compatibility across diverse neural operators without any modification on their structures. Accordingly, operator-agnostic iMOOE can be easily extended to forecast 3D turbulent flows or PDE dynamics on irregular geometries by integrating specialized neural operators that can handle 3D PDEs or irregular spatial domains.
>
> - **Application to 3D fluid dynamics forecasting.** In **Appendix E.2.2 and Table 15 of revised paper**, we have demonstrated that when designating FNO3d [1] as operator experts, iMOOE3d can attain 17.36\% and 39.26\% decrease for nMSE and fRMSE on OOD performance of 3D compressible Navier-Stokes dynamics forecasting. This 3D NS dataset is borrowed from PDEBench [2], and we draw hear and bulk viscosity coefficients from a wide range of values to cover both laminar and turbulent flows.
> - **Application to PDE dynamics on irregular geometries.** In **Appendix E.2.4 and Table 17 of revised paper**, we supplement experiments on OOD forecasting of compressible flows around airfoil. This Airfoil dataset on irregular geometries is proposed in Section 4.1 of OFormer paper [3], where the spatial grid is divided by highly irregular triangular meshes. Each PDE trajectory in training or testing set has a different Mach number and angle of attack. The task is to forecast future 22 steps velocity, density and pressure using their past 4 steps observations. We can easily enable iMOOE to perform PDE forecasting on irregular meshes by designating specialized neural operators on general geometries like OFormer [3], Geo-FNO [4] and VCNeF [5] as operator experts. In the following table, we demonstrate when equipped with the proposed iMOOE framework, three neural operators on irregular geometries can overall attain 12.89\%, 12.01\% decrease on nMSE, fRMSE, exhibiting better OOD results on forecasting flows around Airfoil. We also randomly showcase a flow velocity sequence on triangular meshes in **Figure 15, 16 of revised paper**.
>
> | Operators | Variants | nMSE | RMSE |
> | :---: | :---: | :---: | :---: |
> | VCNeF | Naive | 5.45e-2 | 2.22e-1 |
> |  | **+iMOOE** | **5.08e-2** | **1.98e-1** |
> | Geo-FNO | Naive | 5.15e-2 | 2.16e-1 |
> |  | **+iMOOE** | **4.35e-2** | **1.88e-1** |
> | OFormer | Naive | 4.95e-2 | 2.04e-1 |
> |  | **+iMOOE** | **4.14e-2** | **1.79e-1** |

---

> ### Author Response · Authors · 2025-11-20
> **Response to Reviewer adhf (Part 2)**
>
> **W2-1: Lacking more rigorous justification for two-level PDE invariance.**
>
> Firstly, in the field of PDE learning, the commonly utilized assumption is that many specialized PDE systems demonstrate compositional structures and contains a group of distinct physical processes, as claimed in Section 4.1 of [11]. For example, Navier-Stokes systems are composed of convection, pressure and viscosity, wave systems are composed of Hamiltonian flow and dissipation. This PDE compositionality assumption can be formulated as [6]:
> $\partial _{t}\mathbf{u}(\mathbf{x})=F\left (\sigma _{1}(\mathbf{u},\mathbf{x}),...,\sigma _{i}(\mathbf{u},\mathbf{x}),...,\sigma _{K}(\mathbf{u},\mathbf{x}) \right )$.
> $F(\cdot)$ is a nonlinear function, which represents the compositional relationship among a group of basic operators:
> $\{ \sigma _{i}(\mathbf{u},\mathbf{x})\} _{i=1}^{K}$.
> Such compositional operator structure guides us to design the two-level PDE invariance principle. The proposed iMOOE leverages a fusion network to approximate $F(\cdot)$ and a set of neural operator experts to approximate $\{ \sigma _{i}(\mathbf{u},\mathbf{x}) \} _{i=1}^{K}$.
>
> Secondly, as observed in classical PDE discovery literature such as SINDy [7] and PDE-FIND [8], PDE systems of interests can be naturally represented in a few spatial operators. These PDE discovery methods resort to SVD-based linear combination of a few terms in the fixed operator library. This indeed motivates us to design the operator invariance principle, and also justifies the compositional structure of operator learning.
>
> Thirdly, we justify the composition of neural operators (i.e. iMOOE's architecture design) can approximate arbitrary continuous operators. This is indeed served as critical part of proofs according to the neural operator's universal approximation theorem shown in [9] (Theorem 11 and Lemma 27-28). As illustrated in Section 9.3 and Figure 16 in [9], neural operators that can approximate continuous operators should possess three modules: i) Mapping input to the first hidden representation by a local operator. ii) Mapping each hidden representation to next by a non-local integral kernel operator. iii) Mapping the last hidden representation to output by a local operator. In practical implementation, i) and iii) can be realized by a local neural network like MLP, while ii) should be realized by a global neural network like transformer or FNO layers. The proposed composition of neural operators realizes i) and ii) by concatenating the last hidden representations of multiple neural operators, and realizes iii) by the fusion network. Hence, iMOOE architecture design can also satisfy the universal approximation theorem of operators. In the revised manuscript, we will add a clarification and draw the theoretical connection between iMOOE and standard neural operators.
>
> **W2-2: Connection between invariance principle and optimization objective can be better substantiated.**
>
> In terms of invariance property in Section 2.2, we are motivated by the pioneering work REx [10] on invariant learning for multi-domain generalization. In Section 2.2 of REx [10], it has defined that an invariant predictor which serves as an optimal solution of the min-max OOD generalization problem given in Equation 5 should work equally well across all different domains, i.e. the risks of the developed predictors across domains should be statistically equal. Then, the risk equality principle [10] is proposed in Section 3 and implemented in Equation 8: we can enforce a neural network to realize OOD generalization by simultaneously minimizing the risk of each training domain and the variance of multi-domain training risks. Besides, in Section 3.2 and Appendix E.1 of [10], it has been proved that equalizing risks can aid to learn the true invariant predictor.
>
> To achieve such technique in OOD dynamics forecasting, we define the PDE invariance and use the variance of prediction risks (i.e. prediction error of PDE trajectories) as a regularizer, as presented in **Section 3.2 and Equation 7 of revised paper**. As shown in **Table 3 of revised paper**, we can see that when equipped with the risk equality loss, not only the average prediction errors across 8 test environments decrease, the variance of these multi-environment prediction errors also notably decrease.

---

> ### Author Response · Authors · 2025-11-20
> **Response to Reviewer adhf (Part 3)**
>
> [1] Fourier Neural Operator for Parametric Partial Differential Equations, ICLR 2021.
>
> [2] PDEBENCH: An Extensive Benchmark for Scientific
> Machine Learning, NeurIPS 2022.
>
> [3] Transformer for Partial Differential Equations’ Operator Learning, Transactions on Machine Learning Research, 2023.
>
> [4] Fourier neural operator with learned deformations for pdes on general geometries, Journal of Machine Learning Research, 2023.
>
> [5] Vectorized Conditional Neural Fields: A Framework for Solving Time-dependent Parametric Partial Differential Equations, ICML 2024.
>
> [6] Toward Efficient Kernel-Based Solvers for Nonlinear PDEs, ICML 2025.
>
> [7] Discovering governing equations from data by sparse identification of nonlinear dynamical systems. Proceedings of the national academy of sciences, 2016.
>
> [8] Data-driven discovery of partial differential equations, Science advances, 2017.
>
> [9] Neural Operator: Learning Maps Between Function Spaces With Applications to PDEs, JMLR, 2023.
>
> [10] Out-of-distribution generalization via risk extrapolation (rex), ICML 2021.
>
> [11] Multiple Physics Pretraining for Spatiotemporal Surrogate Models, NeurIPS 2024.

---

> > ### Comment · Reviewer_adhf · 2025-11-27
> >
> > I thank the authors for their detailed and clear responses. Most of my concerns have been addressed and my evaluation remains unchanged.

---

> > > ### Author Response · Authors · 2025-11-27
> > > **Response to Reviewer adhf**
> > >
> > > We greatly appreciate for your careful reading on our response and positive support to our work. If you have any questions or advice regarding iMOOE, we are delighted to have further discussions with you. Thanks for your time and effort!

---

### Official Review · Reviewer_RXdg · 2025-11-01

**Soundness:** 3
**Presentation:** 2
**Contribution:** 2
**Rating:** 4
**Confidence:** 4

**Summary:**

This work is motivated by invariant learning theory and addresses the challenge of out-of-distribution (OOD) prediction, covering both temporal and parametric extrapolation tasks. The key idea is that partial differential equations (PDEs) possess two fundamental types of invariance:
* Operator invariance: individual physical operators (e.g., diffusion, advection, reaction) remain consistent across domains.
* Compositionality invariance: the relationships between these operators and external factors such as physical parameters and forcing terms remain fixed for a given PDE.
To capture these invariances, the authors propose iMOOE (Invariant Mixture of Operator Experts). The method first decomposes a PDE into a set of compositional operators, then uses a Mixture of Operator Experts architecture to model both the invariant operators and their compositional relationships.
The effectiveness of iMOOE is demonstrated across five canonical PDE systems, including diverse parametric and temporal OOD scenarios, as well as a real-world Sea Surface Temperature (SST) dataset. Across all tests, iMOOE consistently outperforms baseline approaches (such as CoDA, GEPS, CAPE, CNO, DPOT, and VCNeF), achieving state-of-the-art performance in challenging OOD prediction tasks.

**Strengths:**

1. The paper presents a framework for improving OOD generalization in PDE simulation, motivated by invariant learning theory.
2. Several test cases are provided, including both temporal and parametric OOD settings, validating the method’s potential on challenging extrapolation problems.
3. The paper includes comprehensive experiments, systematic comparisons with strong baselines, and sensitivity analyses (e.g., number of experts, loss weighting, domain partitioning).

**Weaknesses:**

1. The method assumes that PDEs can be decomposed into known invariant operators. In cases where the governing equations are unknown or highly complex, this assumption may not hold, reducing the method’s applicability.
2. The paper claims “zero-shot OOD generalization”, but the evaluated OOD shifts are primarily controlled parametric variations. It is unclear whether this corresponds to zero-shot generalization to entirely new physical equations.
3. The method’s reliance on multiple neural operator experts increases computational cost, especially for high-dimensional PDEs.

**Questions:**

1. How does iMOOE perform for irregular spatial domains or unstructured meshes? Can the operator decomposition and masking strategy be naturally extended to such cases?
2. Can the authors comment on the generalization between dimensions — for example, if trained on 2D heat conduction, how well would the approach extrapolate to 3D?
3. Beyond parametric shifts, can iMOOE handle structurally different PDEs (e.g., transitioning from Burgers to Navier–Stokes), or does operator invariance only hold within a single PDE family?
4. How sensitive is the method to the diversity of training domains? Is there a minimum level of variation required to achieve robust generalization?

---

> ### Author Response · Authors · 2025-11-20
> **Response to Reviewer RXdg (Part 1)**
>
> *We sincerely appreciate for your careful review and constructive suggestions. Hope our response can address your questions and concerns.*
>
> **W1: The decomposition assumption may not hold when governing PDEs are unknown or highly complex.**
>
> In **Section 2.1 of revised paper**, we have claimed the studied PDE systems in this work are totally unknown. Extensive OOD generalization results have demonstrated that the pure data-driven iMOOE can capture the unknown PDE law by the proposed physics-informed invariant learning. We follow the common assumption in previous PDE learning literature of which PDE systems of interests can be decomposed into a few invariant operators. Please refer to **W2-1 response to Reviewer adhf** for more formal and rigorous justification of the proposed PDE invariance principle. In **Section 4.2, 4.4 and Appendix E.2.2, E.2.4 of revised paper**, we demonstrate iMOOE can perform zero-shot OOD forecasting on a wide range of unknown PDE systems.
>
> **W2\&Q3: It is unclear whether iMOOE can perform zero-shot generalization to entirely new PDEs beyond parametric shifts.**
>
> In this work, we follow the OOD setting defined in the literature, and comply to the hierarchy of generalization for PDEs defined in Section 2 of [7]. In **Section 4 and Appendix C of revised paper**, we have demonstrated iMOOE can tackle various OOD generalization scenarios, including the distribution shifts of initial conditions, forcing functions, physical parameters and temporal domains.
>
> The distinct advantage of iMOOE is its excellent zero-shot OOD capability, while previous PDE forecasting methods still need sufficient test-time samples to achieve satisfactory OOD robustness. This is because invariance-aligned iMOOE should hold in the single family of PDEs due to: i) There are actually multiple ways to decompose a PDE into several operators. The pure data-driven iMOOE will adaptively learn an independent operator splitting way from the provided PDE data. ii) When the PDE structure, i.e. compositionality invariance is changed, iMOOE need new PDE trajectories to retrain the fusion network, which breaks the focused zero-shot OOD setting. Besides, the problem of cross-PDE generalization is quite challenging for current PDE learning methods. For example, the large-scale pretrained PDE foundation models like POSEIDON [8] still require enough samples simulated on unseen PDEs to achieve satisfactory forecasting accuracy, as shown in Appendix D.4 of [8]. We can pursue the zero-shot cross-PDE generalization and proceed to expand the universality of PDE learning methods in future work.
>
> **W3: Reliance on multiple neural operator experts increase computational cost.**
>
> In **Table 1-5 of revised paper**, we have demonstrated that iMOOE with only two FNO experts can prominently outperform other baseline models on zero-shot OOD forecasting. In **Appendix E.8 and Table 21 of revised paper**, we have compared the inference time of FNO-iMOOE with other baseline models, where iMOOE containing two FNOs does not increase the computational overhead to a large extent. We can reduce the computational cost of iMOOE by integrating parameter-efficient neural operators. We believe iMOOD's strong generalization and performance improvements outweigh the mild computation overhead.

---

> ### Author Response · Authors · 2025-11-20
> **Response to Reviewer RXdg (Part 2)**
>
> **Q1: iMOOE's performance on irregular spatial domains or unstructured meshes.**
>
> We leverage the Airfoil forecasting benchmark on non-uniform grid proposed in Section 4.1 of OFormer paper [1], where the underlying grid is divided by highly irregular triangular meshes. The governing 2D compressive flow equation is specified in Appendix D.2. This Airfoil dataset contains 1000 training and 100 testing sequences with different inflow speed (Mach number) and angles of attack. The objective is to jointly predict future 22 steps velocity, density and pressure using past 4 steps states.
>
> One of the key advantages of iMOOE is the elegant compatibility across various neural operators. Hence, iMOOE (including operator splitting-informed design and masking scheme) can integrate neural operators that are able to handle irregular spatial domains or unstructured meshes as well, such as OFormer [1], Geo-FNO [2] and VCNeF [3]. We can easily enable iMOOE on irregular geometries by two adaptations in a flexible manner: i) The prior spatial derivatives on irregular grid are calculated by the finite element method. ii) The frequency enrichment loss relying on fast Fourier transform on uniform grid is discarded. In the following table, it is apparent that when equipped with the proposed iMOOE method, three neural operators can overall realize 12.89\%, 12.01\% decrease on nMSE, fRMSE, exhibiting better OOD forecasting accuracy on flows around Airfoil. We have included such results in **Appendix E.2.4 and Table 17 of revised paper**. We also randomly showcase a flow velocity trajectory in **Figure 15, 16 of revised paper**.
>
> | Operators | Variants | nMSE | RMSE |
> | :---: | :---: | :---: | :---: |
> | VCNeF | Naive | 5.45e-2 | 2.22e-1 |
> |  | **+iMOOE** | **5.08e-2** | **1.98e-1** |
> | Geo-FNO | Naive | 5.15e-2 | 2.16e-1 |
> |  | **+iMOOE** | **4.35e-2** | **1.88e-1** |
> | OFormer | Naive | 4.95e-2 | 2.04e-1 |
> |  | **+iMOOE** | **4.14e-2** | **1.79e-1** |
>
>
> **Q2: Comment on the generalization between dimensions, such as extrapolating from 2D to 3D?**
>
> The general neural operators trained on 2D spatial domain can not meaningfully extrapolate to 3D domain, since the dimensionality of integration area for the learnable non-local integral kernel operator is changed, as stated in Section 3 of [5]. iMOOE can perform zero-shot OOD forecasting for 3D PDEs by integrating dedicated 3D neural operators like FNO3d [6]. We have validated the performance of FNO3d-iMOOE on a 3D NS dataset in **Appendix E.2.2 and Table 15 of revised paper**.
>
> **Q4: Minimum level of training domain diversity required to achieve robust generalization.**
>
> In **Appendix E.7 and Figure 6(b),7(b) of revised paper**, we have illustrated iMOOE can benefit from more diverse training environments on DR forecasting. Such PDE data diversity refers to the changing behaviors of state variables over the spatiotemporal domain are distinct under different physical environments. We utilize the Variance of Spectral Entropy (VSE) metric to quantify the diversity of multi-environment PDE trajectories. According to [4], Spectral Entropy (SE) can represent the unique distributional properties of frequency modes for each spatiotemporal PDE trajectory. SE can be calculated by 3D-FFT. Hence, we can calculate the variance of SE values across multi-environment PDE sequences to represent the diversity of PDE training data. In the following table, we report the VSE of two physical variables in multi-environment DR dataset. We can find that DR training data with larger VSE values (i.e. better diversity) can achieve better OOD results in **Figure 6(b),7(b) of revised paper**. As 32 training environments can already realize acceptable OOD performance, we can empirically claim that when the budget of DR training trajectories is fixed to 1024, the minimal diversity level required by PDE invariance learning can be represented by $VSE_{min}=1.1885$. We can generalize such empirical analysis on training data properties to other PDE datasets.
>
> | Number of DR training environments | u\_VSE | v\_VSE | mean\_VSE |
> | :---: | :---: | :---: | :---: |
> | 4 | 1.0538 | 0.9674 | 1.0106 |
> | 8 | 1.2039 | 1.0824 | 1.1432 |
> | 16 | 1.2770 | 1.0858 | 1.1814 |
> | 32 | 1.2854 | 1.0915 | 1.1885 |
> | 64 | 1.2951 | 1.1300 | 1.2125 |
> | 128 | 1.3006 | 1.1483 | 1.2244 |
> | 256 | 1.2943 | 1.1164 | 1.2054 |
> | 512 | 1.2997 | 1.1431 | 1.2214 |

---

> ### Author Response · Authors · 2025-11-20
> **Response to Reviewer RXdg (Part 3)**
>
> [1] Transformer for Partial Differential Equations’ Operator Learning, Transactions on Machine Learning Research, 2023.
>
> [2] Fourier neural operator with learned deformations for pdes on general geometries, Journal of Machine Learning Research, 2023.
>
> [3] Vectorized Conditional Neural Fields: A Framework for Solving Time-dependent Parametric Partial Differential Equations, ICML 2024.
>
> [4] No-reference image quality assessment based on spatial and spectral entropies, Signal processing: Image communication, 2014.
>
> [5] Neural operator: Learning maps between function spaces with applications to pdes, Journal of Machine Learning Research, 2023.
>
> [6] Fourier Neural Operator for Parametric Partial Differential Equations, ICLR 2021.
>
> [7] Does in-context operator learning generalize to domain-shifted settings?, The symbiosis of deep learning and differential equations, 2023.
>
> [8] Poseidon: Efficient foundation models for pdes, NeurIPS 2024.

---

> > ### Comment · Reviewer_RXdg · 2025-11-25
> >
> > Thanks for providing additional experiments and answering in detail. I have raised my score.

---

> > > ### Author Response · Authors · 2025-11-26
> > > **Response to Reviewer RXdg**
> > >
> > > Dear Reviewer RXdg,
> > >
> > > We sincerely appreciate for your careful reading on our response and positive support to the proposed iMOOE method. If you have further questions or suggestions on our work, welcome to discuss! Thank you very much!

---

### Author Response · Authors · 2025-12-01
**Summary of Discussions**

Dear AC,

We sincerely appreciate you and all the reviewers for your meticulous labor and valuable comments during the discussion period. We also appreciate reviewers' time and efforts in fully evaluating our manuscript. Below, we provide a concise summary of the main revisions we made, along with author-reviewer discussions which has taken place so far.

First of all, it is encouraging to know that in initial reviews, all reviewers recognized the significance of the zero-shot OOD generalization problem for PDE dynamics forecasting, as well as the novelty of our proposed physics-guided invariant learning framework iMOOE based on the two-level PDE invariance principle. We believe our efforts during rebuttal phases have addressed all the concerns and questions raised by the reviewers. Here is the outline of major clarifications during rebuttal phase:

- **Rigorous justification for the proposed two-level PDE invariance principle.** We formally justify the proposed operator invariance and compositionality invariance can hold across many specialized PDE systems from three perspectives.
- **Broad applicability to diverse PDE systems.** We provide additional zero-shot OOD generalization results on PDE systems with irregular or 3D spatial domains, as well as real-world PDE dynamics with sensory measurement noise.
- **Additional in-depth analysis on iMOOE's key designs.** We provide additional, more comprehensive analysis on the effect of several modules, including minimum diversity level of training domains required by invariant learning, robustness of frequency-augmented loss to measurement noise, and interpretation of what the specialized operator experts actually represent.

Secondly, we outline all reviewers' comments during the normal double-blind discussion period (as evidenced by the time of "Official Comment by Reviewer") as follows:

- Reviewer RXdg **fully accepted our rebuttal without any dissent or further questions** and **raised the rating score from 4 to 6**.
- Reviewer adhf claimed most of concerns have been addressed and **kept the positive rating score at 6**.
- Reviewer K9mK did **not replied to our response** before the reviewer system was closed.
- Reviewer YiSh **maintained the positive rating score at 6 and increased confidence from 3 to 4** after one more round of discussion.

In conclusion, our proposed iMOOE is novel in architecture, training paradigms, and comprehensive and strong empirical results. And the **key innovation of iMOOE** is that in order to achieve zero-shot generalizable PDE dynamics forecasting, we devised a physics-informed mixture of operator experts architecture with multi-environment invariant learning which can capture the proposed two-level PDE invariance principle. The **key advantages of iMOOE** lie in its zero-shot generalizability to various unseen OOD forecasting scenarios and seamless compatibility with existing neural operators.

We warmly welcome the Area Chair to raise any further questions, especially for any points we have not addressed previously. We deeply thanks for your time and effort.

---

### Meta-Review · Area_Chair_Putb · 2026-01-08

**Summary:**

This paper addresses the very difficult problem of zero-shot out-of-distribution (OOD) generalization for PDE forecasting.
The authors propose a framework called iMOOE, which is combining a mixture of operator experts
 with a physics-guided invariance principle.

The main reason for my positive recommendation is the very convincing evolution
 of the scores during the rebuttal phase. Pre-rebuttals, the reviewers were quite divided. Several experts, notably Reviewer RXdg and Reviewer K9mK,   had some strong doubts concerning the theoretical foundations and   if the model can scale to more complex, real-world geometries.

However, the authors' response was  solid. To answer to these concerns, they did
not only give some theoretical arguments, but they provided intensive new experiments that were missing in the first version:

1. Scaling to 3D: They have demonstrated the model's effectiveness on 3D Navier-Stokes datasets,
showing a clear reduction of the error.
2. Irregular Domains: They showed that iMOOE works on "Airfoil" meshes,
which proves the method is not limited to simple grids.

Reviewer RXd, who started with a 4 (Marginally Below), was sufficiently impressed by the new
3D results and the clarifications to raise their score.
Also, Reviewer YiSh confirmed his positive position by increasing his confidence level,
noting that the interpretability of the experts was now well-justified.

Even if Reviewer K9mK did not update his score, the authors
have, I believe, addressed the technical points about noise and mathematical rigor with
enough details to satisfy the rest of the reviewers.

In conclusion, the trajectory of this paper during the review process shows a nice paper.

**Reviewer Concerns:**

adressed:
* irregular domain
* scaling to 3d

**Reviewer Scores:**

I am not able to answer this.

---

### Decision · Program_Chairs · 2026-01-26

Accept (Poster)